# Newly developed aircraft routing options for air traffic simulation in the chemistry-climate model EMAC 2.53: AirTraf 2.0

Hiroshi Yamashita[1], Feijia Yin[2], Volker Grewe[1,2], Patrick Jöckel[1], Sigrun Matthes[1], Bastian Kern[1], Katrin Dahlmann[1], and Christine Frömming[1]

[1]Deutsches Zentrum für Luft- und Raumfahrt, Institut für Physik der Atmosphäre, Oberpfaffenhofen, Germany
[2]also at: Delft University of Technology, Aerospace Engineering, Delft, The Netherlands

**Correspondence:** H. Yamashita (hiroshi.yamashita@dlr.de)

**Abstract.** Aviation contributes to climate change and the climate impact of aviation is expected to increase further. Adaptions of aircraft routings in order to reduce the climate impact are an important climate change mitigation measure. The air traffic simulator AirTraf, as a submodel of the ECHAM/MESSy Atmospheric Chemistry (EMAC) model, enables the evaluation of such measures. For the first version of the submodel AirTraf, we concentrated on the general set-up of the model, including departure and arrival, performance and emissions, and technical aspects such as the parallelization of the aircraft trajectory calculation with only a limited set of optimization possibilities (time and distance). Here, in the second version of AirTraf, we focus on enlarging the objective functions by seven new options to enable assessing operational improvements in many more aspects including economic costs, contrail occurrence and climate impact. We verify that the AirTraf set-up, e.g. in terms of number and choice of design variables for the genetic algorithm, allows finding solutions even with highly structured fields such as contrail occurrence. This is shown by example simulations of the new routing options, including around 100 north-Atlantic flights of an Airbus A330 aircraft for a typical winter day. The results clearly show that AirTraf 2.0 can find the different families of optimum flight trajectories (three-dimensional) for specific routing options; those trajectories minimize the corresponding objective functions successfully. The minimum cost option lies between the minimum time and the minimum fuel options. Thus, aircraft operating costs are minimized by taking the best compromise between flight time and fuel use. The aircraft routings for contrail avoidance and minimum climate impact reduce the potential climate impact, which is estimated by using algorithmic Climate Change Functions, whereas these two routings increase the aircraft operating costs. A trade-off between the aircraft operating costs and the climate impact is confirmed. The simulation results are compared with literature data and the consistency of the submodel AirTraf 2.0 is verified.

## 1 Introduction

Climate impact due to aviation emissions is an important issue. Nowadays the global aviation contributes only about 5 % to the anthropogenic climate impact (Skeie et al., 2009; Lee et al., 2009, 2010). However, the aviation's contribution to climate impact is expected to increase further, because global air traffic strongly grows in terms of Revenue Passenger Kilometres (RPK) by 7.4 % in 2016 compared to 2015 (ICAO, 2017). The aviation climate impact consists of carbon dioxide ($CO_2$)

emissions and of non-CO$_2$ effects. The non-CO$_2$ effects comprise nitrogen oxides (NO$_x$) leading to concentration changes of ozone and methane, water vapor (H$_2$O), hydrocarbons (HC), carbon monoxide (CO), sulfur oxides (SO$_x$), non-volatile particulate matter such as black carbon (BC), persistent linear contrails, and contrail-induced cirrus clouds (Wuebbles et al., 2007; Lee et al., 2009; Brasseur et al., 2016). These effects change the radiative balance of the Earth's climate system and cause

radiative impact. The radiative impact potentially drives the climate system into a new state of equilibrium through temperature changes. Lee et al. (2009) stated that the CO$_2$ emission has the main impact and that the estimated radiative forcing (RF) of aviation CO$_2$ in 2005 was 28.0 mWm$^{-2}$ (15.2$-$40.8 mWm$^{-2}$, 90 % likelihood range). The non-CO$_2$ emissions and the induced clouds also have a large effect on RFs; for example, the estimated RFs in 2005 for total NO$_x$ and for persistent linear contrails were 12.6 mWm$^{-2}$ (3.8$-$15.7 mWm$^{-2}$, 90 % likelihood range) and 11.8 mWm$^{-2}$ (5.4$-$25.6 mWm$^{-2}$, 90 %

likelihood range), respectively (Lee et al., 2009). In particular, the radiative impact of contrails remains uncertain and recent studies report higher RF. Burkhardt and Kärcher (2011) estimated the contrail cirrus RF of 37.5 mWm$^{-2}$ for the year 2002; Schumann et al. (2015) reported the RF of 63 mWm$^{-2}$ for the year 2006; and Bock and Burkhardt (2016) estimated the RF of 56 mWm$^{-2}$ for the year 2006. As for time scales of their impacts, the emitted CO$_2$ becomes uniformly mixed in the whole atmosphere and its perturbation remains for millennia. In contrast, the non-CO$_2$ effects occur on short time scales, e.g., the

emitted NO$_x$ remains for a few days to months; the contrails last several hours. Thus, the non-CO$_2$ effects depend strongly on the ambient (local) atmospheric conditions (Fichter et al., 2005; Mannstein et al., 2005; Gauss et al., 2006; Grewe and Stenke, 2008; Frömming et al., 2012; Brasseur et al., 2016; Lund et al., 2017). To investigate measures for reducing the aviation climate impact, the impact of both, CO$_2$ and non-CO$_2$ effects, must be considered; therefore, geographic location, altitude, the time of released non-CO$_2$ emissions and induced clouds, and corresponding local atmospheric conditions need to be considered.

In recent years, Grewe et al. (2017a, b) and Matthes et al. (2012, 2017) have proposed a climate-optimized routing as an important operational measure for reducing the aviation climate impact. This routing allows a significant reduction of the climate impact by optimizing flight routes to avoid regions, where released emissions (including contrails) have a large climate impact. The climate-optimized routing is immediately applicable to present airline fleets, whereas other, more technological measures (e.g., efficient engines, blended wing-body configurations, and laminar flow controls; Green, 2005) require several

25 years before implementation. Moreover, the routing can be used in addition to the technological measures for reducing the aviation climate impact.

Benefits of the climate-optimized routing have been examined before (Gierens et al., 2008; Schumann et al., 2011; Sridhar et al., 2013; Søvde et al., 2014; Lührs et al., 2016); for example, Frömming et al. (2013) and Grewe et al. (2014b) developed Climate Cost Functions (CCFs) for the climate-optimized routing. They calculated global-average RFs resulting from local unit emis-

30 sions (CO$_2$, NO$_x$, H$_2$O and contrails) over the north-Atlantic for typical weather patterns by using the ECHAM/MESSy Atmospheric Chemistry (EMAC) model (Jöckel et al., 2010, 2016). Those RFs were used to calculate the global and temporal average near-surface temperature response over 20 years, which describe the climate impacts (i.e. future temperature changes) caused by those emissions on a per unit basis. The resulting data set is called the CCFs. The CCFs describe the climate impact which is induced by aviation's CO$_2$ and non-CO$_2$ effects (H$_2$O, ozone, methane, ozone originating from methane changes,

and contrails including the spread into contrail-cirrus); and the CCFs of those effects except CO$_2$ are a function of geographic

location, altitude and time. Because of the long residence time of $CO_2$, its impact is the same regardless of location, altitude and time of emission. The obtained CCFs can be used as a measure of the climate impact of aviation and form the basis for the climate-optimized routing. Grewe et al. (2014a) calculated the CCFs for a winter day and optimized one-day trans-Atlantic air traffic (391 eastbound and 394 westbound flights) using the CCFs in the system for traffic assignment and analysis at macro-scopic level (SAAM; Eurocontrol, 2012). They reported that the climate impact decreased by up to 25 % with a small increase in economic costs of less than 0.5 %. This revealed a great potential for the climate-optimized routing. On the other hand, a trade-off between climate impact and economic cost existed, i.e., the climate-optimized and the cost-optimized routings were conflicting strategies. Grewe et al. (2017b) extended this study and investigated the feasibility of the climate-optimized routing for realistic conditions. Similar trans-Atlantic air traffic simulations (about 800 flights) were performed for five representative winter and three representative summer days, taking safety aspects into account. They found that a decrease in potential climate impact of 10 % was achieved by a cost increase of only 1 %.

The benefits of the climate-optimized routing were investigated by using different climate metrics. Ng et al. (2014) optimized flight trajectories for a total climate cost which was calculated by the absolute global temperature change potential (pulse AGTP values for three time horizons; Shine et al., 2005) due to $CO_2$ emission and contrails. A total of 960 trans-Atlantic flights (482 eastbound and 478 westbound flights) was analyzed for a specific summer day. They reported that the climate-optimized routing reduced the total AGTP (for the medium-term climate goal of 50 years) by 38 % with an additional flight time of 3.1 % and with extra fuel use of 3.1 % for the eastbound flights, whereas the routing reduced the total AGTP by 20 % with an additional flight time of 3.0 % and with extra fuel use of 3.7 % for the westbound flights. Generally, aircraft operating costs depend on time and on fuel. Thus, those results indicate the aforementioned trade-off between climate impact and economic cost; this trade-off was also found for the short-term (25 years) and long-term (100 years) climate goals. Grewe et al. (2014a) compared the trade-off between economic costs and climate impact from the one-day trans-Atlantic air traffic simulations described above with respect to three climate metrics: the average temperature response with future increasing emissions (F-ATR20) and the absolute global warming potential with pulse emissions at a 20 year time horizon (P-AGWP20) for short-term climate impacts, and P-AGWP100 (time horizon of 100 years) for long-term climate impacts. The trade-offs obtained with the three metrics were very similar. Although many studies show the benefit of the climate-optimized routing, this routing is not used for the today's flight planning: today's aircraft routing focuses on minimum economic cost. However, if additional costs, such as environmental taxes, for aviation climate impact of $CO_2$ and non-$CO_2$ effects are included in the operating costs, a cost increase due to the climate-optimized routing is possibly compensated (Grewe et al., 2017b). This inclusion can change the current routing strategy, and incentivize airlines to introduce a climate-optimized flight planning.

Here, we present an air traffic simulation model which serves as a basis for the following ultimate two aims: to investigate an eco-efficient aircraft routing strategy that reduces the climate impact of global air traffic over the next few decades, and to estimate its mitigation gain for different aircraft routing strategies. For these aims, the submodel AirTraf (version 1.0) has been developed as one of the submodels of EMAC (Yamashita et al., 2015, 2016). AirTraf can simulate global air traffic in EMAC (online) for various aircraft routing strategies (options). Every flight trajectory is optimized for a selected routing option under

daily changing local atmospheric conditions. AirTraf can take into account where and when aviation emissions are released or contrails form. The road map for our overall study has been shown elsewhere (Grewe et al., 2017b; Matthes et al., 2017).

This paper presents a technical description of the new version of the submodel AirTraf 2.0. The simple aircraft routing options of great circle (minimum flight distance) and flight time (minimum time) were developed in the previous version of AirTraf 1.0 (Yamashita et al., 2016). In AirTraf 2.0, seven new aircraft routing options have been introduced: fuel use, $NO_x$ emission, $H_2O$ emission, contrail formation, simple operating cost (SOC), cash operating cost (COC), and climate impact estimated by the algorithmic Climate Change Functions (aCCFs; Van Manen, 2017; Yin et al., 2018b; Van Manen and Grewe, 2019; Yin et al. (manuscript in preparation, 2020); the Climate Change Functions were previously referred to as the Climate Cost Functions mentioned above). These options represent the objects to be minimized. Overall the nine options have been integrated in AirTraf 2.0, which enable air traffic simulations for the ultimate aims of our study (hereinafter the aircraft routing options are referred to simply as, e.g. the "fuel option"). Thus, the development described in this paper is an indispensable update. Moreover, this paper provides example applications of AirTraf 2.0. Some simulations of the nine routing options were carried out for trans-Atlantic routes for a typical winter day. Optimum flight trajectories and characteristics of the routing options were analyzed.

Here, we mention the importance of the variety of the routing options. Various routing options have been made available in AirTraf 2.0, because not only the climate and the cost options, but also the other options are important subjects for air traffic routing studies. The time option is useful for delay recovery. Because delays cause costs to airlines, pilots are often forced to temporarily use the time option during a flight to maintain flight schedules, although the use of this option increases fuel costs (Cook et al., 2009). The $NO_x$ (Mulder and Ruijgrok, 2008) and contrail options (Fichter et al., 2005; Mannstein et al., 2005; Gierens et al., 2008; Sridhar et al., 2011; Schumann et al., 2011; Rosenow et al., 2017) have been examined as a routing strategy towards climate impact reduction. Moreover, conflicting scenarios (trade-offs) between different routing strategies have been studied; for example, avoiding contrail formation generally increases fuel use and $CO_2$ emissions. Irvine et al. (2014) assessed the trade-off between contrail avoidance and increased $CO_2$ emission ($\sim$ increased fuel use) for a single flight. AirTraf 2.0 enables analyzing those subjects all at once, because all the options are integrated. Normally, one or two specific routing options are available for a flight trajectory optimization in other models. Another aspect to be emphasized compared to other models is that AirTraf performs air traffic simulations not under International Standard Atmospheric (ISA) conditions, not under a fixed atmospheric condition for a specific day, but under comprehensive atmospheric conditions which are calculated by EMAC; that is, AirTraf can simulate air traffic for long-term periods in EMAC, which enables one to examine effects of aircraft routing strategies on climate impact on a long time scale. Last but not least, the aCCFs are new proxies for the climate-optimized routing. An important aim of the AirTraf development is to verify the aCCFs themselves and the routing strategy based on the aCCFs (i.e., the climate option) in multi-annual (long-term) simulations (Yin et al., 2018b).

This paper is organized as follows. Section 2 describes an overview of AirTraf 2.0. Particularly, key changes in the model components are stated. Section 3 presents the results and discussion for the example applications of AirTraf 2.0 using the nine routing options. Section 4 verifies the consistency of the results with literature data. Finally, Sect. 5 concludes this study.

## 2 Overview of AirTraf 2.0

### 2.1 Chemistry-climate model EMAC

The EMAC model is a numerical chemistry and climate simulation system that includes submodels describing tropospheric and middle atmosphere processes and their interaction with oceans, land, and influences coming from anthropogenic emissions (Jöckel et al., 2010, 2016). It uses the second version of the Modular Earth Submodel System (MESSy2) to link multi-institutional computer codes. The core atmospheric model is the 5th generation European Centre Hamburg general circulation model (ECHAM5; Roeckner et al., 2006). For the present study, we applied EMAC (ECHAM5 version 5.3.02 and MESSy version 2.53 updated from the version 2.41 for AirTraf 1.0) in the T42L31ECMWF resolution, i.e. with a spherical truncation of T42 (corresponding to a quadratic Gaussian grid of approximately 2.8° by 2.8° in latitude and longitude) with 31 vertical hybrid pressure levels up to 10 hPa (middle of the uppermost layer). The namelist setup for ECHAM5 simulations (referred to the E5 setup, no chemistry) was employed. Moreover, the submodel AirTraf was coupled to the submodel CONTRAIL (version 1.0; Frömming et al., 2014) for the contrail option, and to the submodel ACCF (version 1.0) for the climate option, using the MESSy interfaces. Further information about MESSy, including the EMAC model system, is available from the MESSy Consortium Website (http://www.messy-interface.org).

### 2.2 Model components of submodel AirTraf

Figure 1 shows the flowchart of the submodel AirTraf 2.0. The present version is based on the model components of AirTraf 1.0, and thus, this section outlines them (updates from AirTraf 1.0 are highlighted in Fig. 1). First, air traffic data and AirTraf parameters are read in the main entry point `messy_initialize` (Fig. 1, dark blue). They consist of a one-day flight plan (including departure and arrival airport pairs, latitude and longitude of the airports, and departure time), Eurocontrol's Base of Aircraft Data (BADA Revision 3.9; Eurocontrol, 2011), ICAO engine performance data (ICAO, 2005), a load factor, jet fuel price, an aircraft routing option, etc. Any arbitrary number of flight plans is applicable and is reused for AirTraf simulations longer than two days. Table 1 lists the relevant data of an A330-301 aircraft and constant parameters used in AirTraf 2.0 (the new parameters are listed in Table 1). Second, all the entries are distributed in parallel by the message passing interface (MPI) standard (called for the main entry point `messy_init_memory`; Fig. 1, blue). Third, the air traffic simulation (called the AirTraf integration; Fig. 1, light blue) is called in the main entry point `messy_global_end`, considering local atmospheric conditions for every flight route. The AirTraf integration uses three modules: the aircraft routing module (Fig. 1, light green), the fuel-emissions-cost-climate calculation module (Fig. 1, light orange), and the flight trajectory optimization module (Fig. 1, dark green). The first module calculates flight trajectories corresponding to a selected routing option. The second module comprises a total energy model based on the BADA methodology (Eurocontrol, 2011; Schaefer, 2012) and the DLR fuel flow method (Deidewig et al., 1996). The third module consists of the Adaptive Range Multi-Objective Genetic Algorithm (ARMOGA version 1.2.0; Sasaki et al., 2002; Sasaki and Obayashi, 2004, 2005). Finally, simulation results are gathered from the MPI tasks. Optimum flight trajectories and global fields of flight properties (four-dimensional Gaussian grid; Fig. 1, rose red) are output. The same assumptions made in AirTraf 1.0 are applied in AirTraf 2.0, e.g., only the cruise flight phase is

considered; trajectory conflicts and operating constraints (e.g., military air space) are neglected. Further details of the model components have been reported by Yamashita et al. (2016).

## 2.3 Calculation procedures of the AirTraf integration

AirTraf 2.0 follows the calculation procedures of AirTraf 1.0 described in detail in Sect. 2.4 of Yamashita et al. (2016). This section reviews the procedures of the AirTraf integration (Fig. 1, light blue) with emphasis on changes by introducing the new routing options.

A one-day flight plan includes departure time for every flight. A flight moves to the flying process (dashed box in Fig. 1, light blue) according to individual departure time in the time loop of EMAC. The flying process comprises four steps: flight trajectory calculation, fuel-emissions-cost-climate calculation, aircraft position calculation, and gathering global emissions (bold-black boxes in Fig. 1, light blue). The first step finds an optimum flight trajectory for a selected routing option by using the aircraft routing module (Fig. 1, light green), in which the seven new routing options are introduced in AirTraf 2.0. The flight trajectory optimization module (Fig. 1, dark green) executes the flight trajectory optimization under atmospheric conditions at the departure day and time of the flight. Thus, the optimum flight trajectory varies day by day. Note that the three-dimensional wind components ($u$, $v$, $w$) are considered in the flight trajectory optimization for all routing options. The resulting optimum flight trajectory consists of waypoints ($i = 1, 2, \cdots, n_{\mathrm{wp}}$) and flight segments ($i = 1, 2, \cdots, n_{\mathrm{wp}} - 1$), where $i$ is the index arranged from the departure ($i = 1$) to the arrival ($i = n_{\mathrm{wp}}$), and $n_{\mathrm{wp}}$ is the number of waypoints (see Fig. 3 of Yamashita et al., 2016). Table 2 lists flight properties calculated for the waypoints, the flight segments, and the whole trajectory. In AirTraf 2.0, 15 new properties are calculated, as highlighted in Table 2.

The second step, which is linked to the fuel-emissions-cost-climate calculation module (Fig. 1, light orange), calculates the flight properties of fuel, $NO_x$ emission, COC, etc. under the atmospheric conditions (Table 2, third group). This calculation is performed once at the departure time of the flight. The methodologies of the fuel-emissions calculation module developed in AirTraf 1.0 are expanded in AirTraf 2.0. Details of the fuel-emissions calculation module and its reliability have been reported in Sects. 2.5, 2.6, and 5 of Yamashita et al. (2016).

The third step moves the aircraft to a new position along the optimum flight trajectory corresponding to the time steps of EMAC, by referring to the estimated time when the aircraft passes through the waypoints (called the estimated time over ETO, Table 2).

At the fourth step, the individual flight properties corresponding to a flight path for one time step of EMAC are gathered into the aforementioned global fields: $NO_x$ emission, $H_2O$ emission, fuel use, flight distance, contrail distance ($PCC_{\mathrm{dist}}$), and average temperature responses for the time horizon of 20 years (ATR20s of ozone, methane, water vapor, $CO_2$, contrails, and total; see Sect. 2.5.7) are gathered along the flight segments (Table 2); the global fields of $PCC_{\mathrm{dist}}$ and ATR20s are newly calculated by AirTraf 2.0. If the aircraft reaches the last waypoint in the time loop of EMAC, the aircraft has landed (i.e., the flight quits) and the flying process ends for this flight.

## 2.4 Flight trajectory optimization

The flight trajectory optimization methodologies described by Yamashita et al. (2016) are also used for the new routing options and are outlined in this section. The flight trajectory optimization module (Fig. 1, dark green) executes the optimization. The module consists of ARMOGA (version 1.2.0; Sasaki et al., 2002; Sasaki and Obayashi, 2004, 2005), which is a stochastic optimization algorithm.

A solution $x$ (the term is synonymous with the flight trajectory) is a vector of $n_{dv}$ design variables: $x = (x_1, x_2, \cdots, x_{n_{dv}})^T$, here $n_{dv} = 11$. With the design variable index $j$ ($j = 1, 2, \cdots, n_{dv}$), $x_j (j = 1, 2, \cdots, 6)$ indicate longitudes and latitudes, and $x_j (j = 7, 8, \cdots, 11)$ indicate altitudes. The $j^{\text{th}}$ design variable varies between lower and upper bounds $[x_j^l, x_j^u]$. The bounds of $[x_j^l, x_j^u]$ ($j = 1, 2, \cdots, 6$) are automatically set for a given airport pair, whereas those of $[x_j^l, x_j^u]$ ($j = 7, 8, \cdots, 11$) are set as $[x_j^l, x_j^u] = [\text{FL290, FL410}]$ (flight levels; FL290 and FL410 denote 29 000 and 41 000 ft, respectively). Geographic locations of the airport pair are set according to the flight plan; altitudes of the airport pair are set to FL290. Given values of $x_j (j = 1, 2, \cdots, n_{dv})$, a three-dimensional flight trajectory is represented by a B-spline curve (third-order) between the airport pair (an illustration is given in Fig. 6 of Yamashita et al., 2016).

The initial population operator (Fig. 1, dark green) generates initial values of $x_j (j = 1, 2, \cdots, n_{dv})$ at random within the lower and upper bounds, and creates an initial "population," which represents a random set of solutions. The population size is set by $n_p$ and ARMOGA starts its search with the solutions. An evaluation function $f$ (called an objective function) is defined, depending on a selected routing option (see Sect. 2.5), and a single-objective optimization problem can be written as follows:

$$\left. \begin{array}{c} \text{Minimize} \ \ f \\ \text{Subject to} \ \ x_j^l \leq x_j \leq x_j^u, \ \ j = 1, 2, \cdots, n_{dv} \end{array} \right\}, \tag{1}$$

where no constraint function is used. The ARMOGA solves the optimization problem by the following genetic operators: evaluation, selection, crossover, and mutation (Fig. 1, dark green; Holand, 1975; Goldberg, 1989). A value of $f$ is calculated for each of the solutions by the evaluation operator. In this study, good solutions were identified in the population by the Fonseca-Fleming Pareto ranking method (Fonseca et al., 1993); the stochastic universal sampling selection (Baker, 1985) was used for the selection operator to pick two solutions (parent solutions) from the population; the Blend crossover operator (BLX-alpha; Eshelman, 1993) was applied to the parent solutions to create new solutions (child solutions); the revised polynomial mutation operator (Deb and Agrawal, 1999) was used to add a disturbance to the child solutions. When those processes are iterated for a number of generations (the term "generation" represents one iteration of ARMOGA; this is set by $n_g$), the population of solutions is improved by reducing $f$, and another superior population is created in subsequent generations. Finally, the ARMOGA finds the best solution (one optimum flight trajectory) with the minimum value of $f$ through the whole generations; the flight properties of the solution are stored, as shown in Table 2. The flight trajectory optimization stated above is executed for every airport pair. Detailed descriptions of the optimization methodologies, appropriate ARMOGA parameter settings, and the accuracy of the optimization module have been presented in Sect. 3.2 of Yamashita et al. (2016).

### 2.5 Formulations of objective functions for new aircraft routing options

In AirTraf 2.0, seven new objective functions were developed for the new aircraft routing options. The following subsections describe formulations of the objective function $f$ for those options. To calculate $f$, the fuel-emissions-cost-climate calculation module (Fig. 1, light orange) is used as necessary by the evaluation operator (Fig. 1, dark green) in the flight trajectory optimization.

#### 2.5.1 Fuel use

The objective function for the fuel option represents the sum of fuel use $\mathrm{kg(fuel)}$ of a flight:

$$f = \sum_{i=1}^{n_{\mathrm{wp}}-1} \mathrm{FUEL}_i, \tag{2}$$

where $\mathrm{FUEL}_i$ is the fuel use of the $i^{\mathrm{th}}$ flight segment (Table 2).

#### 2.5.2 $\mathrm{NO}_x$ emission

The objective function for the $\mathrm{NO}_x$ option represents the sum of $\mathrm{NO}_x$ emission $\mathrm{g(NO_x)}$ of a flight:

$$f = \sum_{i=1}^{n_{\mathrm{wp}}-1} \mathrm{NO}_{x,i} = \sum_{i=1}^{n_{\mathrm{wp}}-1} (\mathrm{FUEL}_i \mathrm{EINO}_{x,a,i}), \tag{3}$$

where $\mathrm{NO}_{x,i}$ is the $\mathrm{NO}_x$ emission of the $i^{\mathrm{th}}$ flight segment; $\mathrm{EINO}_{x,a,i}$ is the $\mathrm{NO}_x$ emission index under actual flight conditions at the $i^{\mathrm{th}}$ waypoint (Table 2) and is calculated using the ICAO engine performance data (ICAO, 2005; see Sect. 2.6 of Yamashita et al., 2016).

#### 2.5.3 $\mathrm{H_2O}$ emission

The objective function for the $\mathrm{H_2O}$ option represents the sum of $\mathrm{H_2O}$ emission $\mathrm{g(H_2O)}$ of a flight:

$$f = \sum_{i=1}^{n_{\mathrm{wp}}-1} \mathrm{H_2O}_i = \mathrm{EIH_2O} \sum_{i=1}^{n_{\mathrm{wp}}-1} \mathrm{FUEL}_i, \tag{4}$$

where $\mathrm{H_2O}_i$ is the $\mathrm{H_2O}$ emission of the $i^{\mathrm{th}}$ flight segment (Table 2); $\mathrm{EIH_2O}$ is the emission index of $\mathrm{H_2O}$ and was set as $\mathrm{EIH_2O} = 1230\ \mathrm{g(H_2O)(kg(fuel))}^{-1}$ (Table 1). The $\mathrm{H_2O}$ emission is proportional to the fuel use by assuming an ideal combustion of jet fuel. Thus, this option yields the same results as the fuel option in AirTraf 2.0. If an alternative fuel option is introduced, the $\mathrm{H_2O}$ option probably differs from the fuel option, because the emission index may not be constant.

#### 2.5.4 Contrail formation

Yin et al. (2018a) developed the routing option to avoid contrail formations by using the submodel CONTRAIL (version 1.0; Frömming et al., 2014), which calculates the potential persistent contrail cirrus coverage $\mathrm{Potcov}$ (Ponater et al., 2002;

Burkhardt et al., 2008; Burkhardt and Kärcher, 2009; Grewe et al., 2014b) within an EMAC grid box. The Potcov represents the fraction of the grid box, which can be maximally covered by contrails under the simulated atmospheric condition. The threshold for contrail formation is determined from a parameterization scheme based on the thermodynamic theory of contrails, i.e., the Schmidt-Appleman theory (Schmidt, 1941; Appleman, 1953; Schumann, 1996). In the CONTRAIL submodel, Potcov indicates the difference between the maximum possible coverage of both, contrails and cirrus, and the coverage of natural cirrus alone; values of Potcov along the waypoints are taken from the nearest grid box (Table 2). With that, we define a contrail distance ($\mathrm{PCC_{dist}}$) in $\mathrm{km(contrail)}$ as Potcov multiplied by the flight distance in $\mathrm{km}$. The corresponding routing option minimizes the total contrail distance of a flight and thus the objective function is formulated as:

$$f = \sum_{i=1}^{n_{\mathrm{wp}}-1} \mathrm{PCC}_{\mathrm{dist},i} = 10^{-3} \sum_{i=1}^{n_{\mathrm{wp}}-1} (\mathrm{Potcov}_i d_i), \tag{5}$$

where $\mathrm{PCC}_{\mathrm{dist},i}$ is the contrail distance of the $i^{\mathrm{th}}$ flight segment; $\mathrm{Potcov}_i$ is the potential persistent contrail cirrus coverage at the $i^{\mathrm{th}}$ waypoint; and $d_i$ is the flight distance of the $i^{\mathrm{th}}$ flight segment (Table 2). Note that the objective function is formulated in the simple form to consider only the contrail distance. Thus, further physical processes such as contrail spreading, changes in contrail coverage area, contrail lifetime, and the contrail radiative forcing are not included.

### 2.5.5 Simple operating cost (SOC)

The cost index (CI) is set during a real flight to manage airline operation costs and is defined as the ratio of time cost to fuel cost (CI = time cost/fuel cost). A low CI value causes an aircraft to minimize fuel use with a sacrifice of flight time, which enables a long-range flight. Conversely, a high CI value causes the aircraft to minimize flight time with an extra fuel use. Generally, the operating costs are a function of flight time and fuel. Thus, the minimum cost solution lies in a trade-off between flight time and fuel (Cook et al., 2009; Marla et al., 2016). Here, the objective function simply represents the sum of the time and the fuel costs on the basis of the CI features:

$$f = \mathrm{SOC} = c_t \sum_{i=1}^{n_{\mathrm{wp}}-1} \frac{d_i}{V_{\mathrm{ground},i}} + c_f \sum_{i=1}^{n_{\mathrm{wp}}-1} \mathrm{FUEL}_i, \tag{6}$$

where $c_t$ and $c_f$ are the unit costs of time and fuel, respectively (Table 1); $V_{\mathrm{ground},i}$ is the ground speed at the $i^{\mathrm{th}}$ waypoint (Table 2). Note that the $c_t$ includes the cost elements for flight crew, cabin crew, and maintenances for both, airframe and engines.

### 2.5.6 Cash operating cost (COC)

The COC is a comprehensive economic criterion for evaluating airline operation costs (Liebeck et al., 1995). The COC includes the cost elements for flight crew, cabin crew, landing fee, navigation fee, fuel, and maintenances for both, airframe and engines (no costs for depreciation, insurance, and interest are included). The COC calculation method for international flights (Liebeck et al., 1995) was employed. Those cost elements were calculated on the basis of the price in 1993 and were scaled to 2015 by the average U.S. inflation rate of average consumer prices $r_{\mathrm{inf}}$ (Table 1; IMF, 2016). Only the fuel cost was

directly calculated with the current jet fuel price JFP (Table 1; IATA, 2017). A block time and a block fuel originally used in the method were replaced by the total flight time FT and the fuel use of $\sum_{i=1}^{n_{\mathrm{wp}}-1} \mathrm{FUEL}_i$ in AirTraf 2.0, respectively (Table 2). The objective function can be written as:

$$f = \mathrm{COC} = C_{\mathrm{flightcrew}} + C_{\mathrm{cabincrew}} + C_{\mathrm{landing}} + C_{\mathrm{navigation}} + C_{\mathrm{fuel}} + C_{\mathrm{airframe}} + C_{\mathrm{engine}}, \tag{7}$$

where $C$ denotes a cost. A detailed description of the COC calculation method has been reported in Liebeck et al. (1995). Given the parameters and variables listed in Tables 1 and 2, Eq. (7) becomes a function of the flight time and the fuel.

### 2.5.7    Climate impact

The climate-optimized routing was carried out by using the aCCFs (Van Manen, 2017; Yin et al., 2018b; Van Manen and Grewe, 2019; Yin et al. (manuscript in preparation, 2020)) calculated by the submodel ACCF. The aCCFs are approximation functions
based on regression analyses for the CCFs data set, which was obtained from detailed EMAC model simulations including radiative impacts (see Sect. 1); the CCFs data set for contrails was exceptionally obtained from contrail RF calculations based on the European Centre for Medium-Range Weather Forecasts (ECMWF) Re-Analysis Interim (ERA-Interim) data (Dee et al., 2011) and contrail trajectory data (Yin et al. (manuscript in preparation, 2020); the definition of the aCCFs is provided in the Appendix and examples are shown in Fig. S1 in the Supplementary material). The aCCFs represent a correlation of meteo-
rological variables at the time of flight with anticipated climate impacts, i.e., ATR20s of ozone, methane, water vapour, $CO_2$, and contrails are estimated on a per unit basis by

$$\mathrm{ATR20}_{\mathrm{O_3},i} = \mathrm{aCCF}_{\mathrm{O_3},i} \times \mathrm{NO}_{x,i} \times 10^{-3}, \tag{8}$$

$$\mathrm{ATR20}_{\mathrm{CH_4},i} = \mathrm{aCCF}_{\mathrm{CH_4},i} \times \mathrm{NO}_{x,i} \times 10^{-3}, \tag{9}$$

$$\mathrm{ATR20}_{\mathrm{H_2O},i} = \mathrm{aCCF}_{\mathrm{H_2O},i} \times \mathrm{FUEL}_i, \tag{10}$$

$$\mathrm{ATR20}_{\mathrm{CO_2},i} = \mathrm{aCCF}_{\mathrm{CO_2}} \times \mathrm{FUEL}_i, \tag{11}$$

$$\mathrm{ATR20}_{\mathrm{contrail},i} = \mathrm{aCCF}_{\mathrm{contrail},i} \times \mathrm{PCC}_{\mathrm{dist},i}, \tag{12}$$

where the respective aCCF values of ozone, methane, water vapour, $CO_2$, and contrails are given as flight properties at the $i^{\mathrm{th}}$ waypoint. These five ATR20s are calculated for flight segments (Table 2) and are combined into an objective function to represent an anticipated climate impact of a flight (in K):

$$\mathrm{ATR20}_{\mathrm{total},i} = \mathrm{ATR20}_{\mathrm{O_3},i} + \mathrm{ATR20}_{\mathrm{CH_4},i} + \mathrm{ATR20}_{\mathrm{H_2O},i} + \mathrm{ATR20}_{\mathrm{CO_2},i} + \mathrm{ATR20}_{\mathrm{contrail},i}, \tag{13}$$

$$f = \sum_{i=1}^{n_{\mathrm{wp}}-1} \mathrm{ATR20}_{\mathrm{total},i}, \tag{14}$$

where $\mathrm{ATR20}_{\mathrm{contrail},i}$ can take positive and negative values, because the $\mathrm{aCCF}_{\mathrm{contrail}}$ consists of two formulas for the day-time and night-time contrail effects (see Eq. (A5) in the Appendix). We acknowledge the large uncertainties in the global temperature response, especially from contrails ($\mathrm{ATR20}_{\mathrm{contrail}}$) due to uncertainties in the efficacy of the contrail forcing

(Hansen et al., 2005; Ponater et al., 2005). In addition, the aCCFs are derived based on the CCFs data of the north-Atlantic region and are applicable to the northern and high latitudes. Further details of the aCCFs have been reported in the literature mentioned above.

## 3 Example application: one-day simulation with new aircraft routing options

### 3.1 Simulation setup

Nine one-day simulations were carried out for a demonstration of AirTraf 2.0. Table 3 lists the simulation setups. The same setups that we used for the consistency check for AirTraf 1.0 simulations (Yamashita et al., 2016) were employed; only the simulation period was changed into a recent day, which showed a typical weather condition in winter with a strong jet stream (see Fig. S2 in the Supplementary material). The flight altitude for the great circle option was set to FL350; the altitude for the other options was calculated in the trajectory optimization within [FL290, FL410], as mentioned in Sect. 2.4. The trans-Atlantic flight plan (103 flights) of an Airbus A330 aircraft was provided by Grewe et al. (2014a) and REACT4C (2014). The setups for the optimization parameters were determined by the benchmark tests (Yamashita et al., 2016).

### 3.2 Optimized flight trajectories and global fields

To display typical simulation outputs, the obtained optimized trajectories and global fields for the contrail, the COC, and the climate options are shown. Figure 2 shows the optimized trajectories for those options (optimized trajectories for other options are shown in Supplement Fig. S3). Obviously, the optimum trajectories vary with the routing options. Figures 2c and 2d show that the COC optimum trajectories of the eastbound flights leap up over the North Atlantic Ocean, whereas the trajectories of the westbound flights are shifted northward. As the jet stream is located at around $50°$W and $40°$N (see Fig. S2 in the Supplementary material), the eastbound trajectories are optimized to benefit from tailwinds of the jet stream and the westbound trajectories avoid headwinds of the jet by detouring northward. In addition, most of those trajectories are located at high flight altitudes ($\sim$FL410, $12.5\,\mathrm{km}$). Figure 3 shows the mean fuel consumption (in $\mathrm{kg(fuel)min^{-1}}$) vs. mean flight altitude (in $\mathrm{km}$) for individual flights for the three routing options. Because fuel consumption decreases as a result of aerodynamic drag reduction at high altitudes (Fichter et al., 2005; Schumann et al., 2011; Yamashita et al., 2016), the COC optimum trajectories select the high flight altitudes, as shown in Fig. 3. We acknowledge that limitations of BADA 3 affect the selection of the flight altitudes (the same applies to the fuel, the $NO_x$, the $H_2O$ and the SOC options; see Fig. S3 in the Supplementary material). According to Nuic et al. (2010), BADA 3 has a tendency to underestimate aircraft fuel consumption at high altitudes and Mach numbers, as the compressibility effect and wave drag are not modeled. These effects will cause differences in the selection of the flight altitudes. In contrast, the contrail and the climate options show complex shaped trajectories with various flight altitude changes (see Figs. 2a, 2b, 2e and 2f).

The global fields of fuel use, contrail distance, and climate impact indicated by $ATR20_{total}$ for the three options are shown in Fig. 4, where distributions represent sum of all the flights during the day. We see from Figs. 4b, 4e and 4h that the contrail

option certainly decreases the contrail formation, which is mostly located over northwest Europe and over the east coast of the U.S. Comparison of Figs. 4a, 4d and 4g shows that the COC option produces a narrower fuel distribution than that of the contrail and climate options. In addition, Figs. 4c, 4f and 4i show that the climate option decreases the positive values of $ATR20_{total}$ (warming effects) over northwest Europe and over the east coast of the U.S., and produces regionally negative values (cooling effects) near Iceland and over eastern Canada, which result in the net climate impact reduction. A comprehesive analysis of the optimized trajectories for the calculated fields is beyond the scope of this paper. However, it is apparent from Fig. 4 that the optimized trajectories successfully decrease the respective objects (target measures) which should be minimized (this point is discussed quantitatively in Sect. 3.3).

### 3.3   Characteristics of aircraft routing options

To examine the characteristics of the routing options, Table 4 lists a summary of nine performance measures of the one-day air traffic (total 103 flights) for specific routing options (bar charts are given in Supplement Fig. S4). Relative changes (in %) to the COC option are also listed in Table 4, considering this option as a reference (the COC option is assumed to be the current aircraft routing strategy). Table 4 shows that individual options successfully minimize their own object (target measure; see measures marked with an asterisk in Table 4). These results confirm that the new routing options work correctly in AirTraf 2.0, since we solve a single-objective minimization problem defined by Eq. (1) for each routing option.

The individual routing options are now discussed in turn. We see from Table 4 that the great circle option has the minimum flight distance of $660.3 \times 10^3$km, whereas this option increases the other measures. The time option shows the minimum flight time of 739.4 h with a large penalty on fuel use, $NO_x$ emission, $H_2O$ emission, SOC, COC, and $ATR20_{total}$ (further discussion in Sect. 4). The fuel option shows the minimum fuel use of 3758.5 ton. Of the nine routing options, the fuel (and also the $H_2O$), the $NO_x$, the SOC, and the COC options obtain similar values on all the measures (see also Supplement Fig. S4): these options show decreased fuel use, $NO_x$ and $H_2O$ emissions, SOC, and COC, whereas contrail distance and $ATR20_{total}$ increase. The difference among these options is considered significant for airline operations and thus is discussed in more detail in Sect. 4. The contrail option shows the minimum contrail distance of $26.3 \times 10^3$km and the second-lowest $ATR20_{total}$ of $3.45 \times 10^{-7}$K, whereas the other measures increase considerably. This option allows aircraft to widely detour the potential contrail regions (because no constraint function is used in Eqs. (1) and (5); see below for more discussion). Thus, the flight distance, the flight time and the fuel use increase drastically, which results in the increase of $NO_x$ and $H_2O$ emissions, SOC, and COC. In particular, the contrail option shows the highest COC of 5.99 Mil.USD of the nine routing options. Comparing the contrail option with the COC option indicates that the contrail distance decreases with an additional fuel use of 8.3 $kg(fuel)(km(contrail))^{-1}$ (i.e., the additional COC of 6.20 $USD(km(contrail))^{-1}$). The SOC and the COC options are comparable. The two options show similar values for all the measures and have the same minimum SOC of 3.96 Mil.USD and COC of 5.35 Mil.USD. In fact, the obtained optimum trajectories for those options are approximately the same (see Figs. 2c, 2d and Supplement Figs. S3k and S3l). This is because the objective function of the two options is a function of flight time and fuel, as defined in Eqs. (6) and (7). An interesting aspect of their performance measures is that both options do not correspond to the minimum flight time and fuel use (see further discussion in Sect. 4). The climate

option achieves the minimum $\text{ATR20}_{\text{total}}$ of $1.96 \times 10^{-7}\text{K}$ and shows the second-shortest contrail distance of $92.6 \times 10^3\text{km}$, whereas the other measures increase, particularly this option shows the second-highest COC of 5.87 Mil.USD. The present results indicate that the contrail and the climate options considerably reduce the climate impact indicated by $\text{ATR20}_{\text{total}}$; however, these options increase COC. The cost-benefit performance (i.e., the COC increment per $\text{ATR20}_{\text{total}}$ reduction) for

the contrail and the climate options are 0.24 and 0.13 $\text{Mil.USD}(10^{-7}\text{K})^{-1}$, respectively. Thus, the climate option seems to be a more cost-effective option. Note that this performance is a narrow result obtained using AirTraf 2.0 under the specific conditions (e.g., the simulations were carried out with the 103 north-Atlantic flights on December 1, 2015, as shown in Table 3). Figure 5 shows the contrail distance (in $10^3\text{km}$) vs. $\text{ATR20}_{\text{contrail}}$ (in $10^{-7}\text{K}$) for individual flights for the contrail, the COC, and the climate options. We see that the contrail option decreases the contrail distance drastically and shows the positive

values of $\text{ATR20}_{\text{contrail}}$ for almost all the flights. On the other hand, the climate option has the longer contrail distances than those of the contrail option (although the climate option achieves the second-shortest total contrail distance, as shown in Table 4) and shows the negative values of $\text{ATR20}_{\text{contrail}}$ for many flights. These results imply that the contrail option minimizes the overall contrail distance at all times, whereas the climate option actively forms cooling contrails during the day and avoids the formation of warming contrails during the day and night. Finally, we believe that the climate benefits described above are most

likely an upper limit, because airspace congestion and air traffic management could reduce the flexibility for flights to perform these trajectory optimizations.

## 4   Discussion: verification of the one-day AirTraf simulation results

This paper presents the extended version of the submodel AirTraf, which offers additional aircraft routing options for defining overall target functions for the flight trajectory optimization. To confirm the consistency of AirTraf simulations, the relative

changes in the performance measures among the routing options (listed in Table 4 in parentheses) are compared with previous studies. The quantitative values of the changes in the performance measures vary, depending on different methodologies, atmospheric conditions, simulation periods, flight plans, aircraft/engine types, cost/climate impact metrics, etc. Thus, a direct comparison in magnitude of our results with published studies is difficult; the sign of the relative changes in the measures is compared. Note that the great circle and the time options have been verified before (Yamashita et al., 2016). In addition, the

$H_2O$ option yields the same results as the fuel option (see Sect. 2.5.3 and Table 4); the SOC option is comparable to the COC option (see Sect. 3.3 and Table 4). Thus, we omit any discussion of the $H_2O$ and the SOC options here.

First, the time, the fuel, and the COC options are analyzed. As defined in Sect. 2.5.6, COC is a combined function of flight time and fuel. To minimize COC, one may attempt to reduce both factors simultaneously; however, a trade-off between the flight time and the fuel generally exists. Table 4 shows that the time penalty of flying minimum fuel trajectories is 2.4 percentage

points (%pt), whereas the fuel penalty of flying minimum time trajectories is 20.3 %pt. A similar trade-off was reported by two published studies. Celis et al. (2014) addressed a single-objective flight trajectory optimization on total flight time and fuel use, respectively, under ISA conditions. A typical single-aisle aircraft (150 passengers) with twin turbofan engines was assumed; the aircraft speed and the flight altitude in eight flight segments were optimized for a given flight trajectory (a quasi-

full flight profile optimization). Compared to the minimum time trajectory, the fuel optimum trajectory decreased the fuel use by 31.7 %pt with increasing flight time by 14.0 %pt. Rosenow and Fricke (2016) compared performances for the minimum time and the minimum fuel trajectories for a flight from Frankfurt (Main) to Dubai for a Boeing B777 freighter on February 2, 2016, at 12 a.m. The comparison showed that the fuel optimum trajectory decreased fuel use by 8.0 % with increasing time by

3.7 %. These studies imply that the minimum COC solution lies between the minimum time and the minimum fuel solutions. In fact, Table 4 shows that the COC option has more flight time than that of the time option, and that the COC option consumes more fuel than that of the fuel option. The COC option yields the values of compromise (i.e. not minimum) of flight time and fuel. Nonetheless, this option achieves the minimum COC. The submodel AirTraf 2.0 can consistently differentiate those three solutions.

To support the discussion above, the fuel and the COC options are compared in detail. Erzberger and Lee (1980) compared the minimum fuel and the minimum direct operating cost (DOC) trajectories for a short-haul route for a Boeing 727-100 aircraft on the basis of optimum control theory (Bryson and Ho, 1969) under U.S. Standard Atmospheric conditions. They showed that flying "minimum fuel" reduced fuel use by 6.9 %, whereas the time and the DOC penalties of the trajectory were 23 and 6 %, respectively (constrained thrust case). Our results in Table 4 show that the fuel option reduces fuel use by 0.1 %, whereas the

time and the COC penalties of the option are 0.1 and 0.03 %, compared to those measures of the COC option. The signs of these relative changes obtained from our results agree with those shown by Erzberger and Lee (1980). In addition, the time and the COC options are compared in a perspective of airline operating economics. Although the time option increases fuel use, $NO_x$ emission, $H_2O$ emission, SOC, COC, and $ATR20_{total}$ (fuel use and COC increase by 20.2 and by 6.1 %, respectively), the option decreases flight time by 2.3 %, compared to that of the COC option. In other words, the time option reduces flight

time with the extra cost of 19 034.74 $USDh^{-1}$ (= 269.66 $EURmin^{-1}$; converted by 1 USD = 0.85 EUR on September 18, 2018 (European Central Bank, 2018)). In a context of delay recovery, this extra cost is the same order of magnitude to flight delay costs. If the flight delay costs exceed the extra cost due to the time option, operators would determine to fly faster by using the time option to recover the delay. Cook et al. (2004, 2009) reported that the flight delay costs, which are associated with delayed passengers, additional fuel use, flight crew, cabin crew, and marginal maintenance costs, reached several hundred

Euros per minute. The extra cost calculated from our results agrees well with this report.

    Compared to the COC option, the $NO_x$ option decreases the $NO_x$ emission by 0.5 %, leading to a COC increase of 0.2 %. Mulder and Ruijgrok (2008) analyzed effects of varying cruise conditions on $NO_x$ emission and on DOC from the cruise $NO_x$ simulation model (Bremmers, 1999) by assuming a cruise range of 5800 km with a Boeing 747-400 aircraft under ISA conditions. They clearly concluded that a reduction of $NO_x$ emission caused a cost increase. Our results agree well with this

conclusion. Moreover, the $NO_x$ option differs from the fuel option, because the amount of $NO_x$ emission depends not only on fuel use, but also on the $NO_x$ emission index, as defined in Eq. (3). The emission index depends strongly on the ambient atmospheric conditions at every waypoint (see Sect. 2.6 of Yamashita et al., 2016). Table 4 shows that the $NO_x$ option decreases the $NO_x$ emission by 0.3 %pt, whereas this option increases flight time by 0.2 %pt and fuel use by 0.2 %pt, compared to those measures of the fuel option. Celis et al. (2014) addressed a single-objective flight trajectory optimization on total fuel use and

on $NO_x$ emission, respectively, with the same simulation setup described above. Compared to the minimum fuel trajectory, the

minimum $NO_x$ trajectory decreased $NO_x$ emission by 10.4 %pt, whereas the trajectory increased time by 1.0 %pt and fuel use by 3.9 %pt. The signs of the relative changes obtained from our results are in good agreement with those shown by Celis et al. (2014).

The contrail option drastically decreases contrail distance by 79.8 % and $ATR20_{total}$ by 43.4 %, whereas this option in-
creases fuel use by 23.0 % and COC by 12.0 %, compared to those measures of the COC option. The contrail option is effective
in order to reduce the climate impact, as pointed out by previous studies introduced in Sect. 1. Here, those relative changes
in the measures are compared with two published studies. Rosenow et al. (2017) performed a one-day European's air traffic
optimization on July 25, 2016. The total number of 13 584 flights over Europe (containing 16 aircraft types) was employed;
their three dimensional flight profiles were optimized for airline costs (termed as the cost performance indicators CPI) and en-
vironmental impacts (termed as the ecological performance indicators EPI). They revealed that an additional contrail avoidance
intent decreased contrail costs by 31.5 % (contrail formations were converted into a monetary value) and EPI by 5.2 %, whereas
the intent increased fuel use by 0.05 % and CPI by 0.5 % over those of the minimum cost strategy. The signs of the relative
changes obtained from our simulations are consistent with those shown by Rosenow et al. (2017). Furthermore, Sridhar et al.
(2013) applied a contrail reducing strategy to aircraft flying between 12 airport pairs (287 flights) in the United States on April
12, 2010. The three-dimensional contrail reducing strategy showed a trade-off between contrail formation time (time spent in
traveling through contrail formation regions) and fuel consumption. Representative points on the trade-off curve showed that
the contrail formation time decreased by 4415 and by 5301 $\mathrm{min}$ with an additional fuel use of 20 000 and of 131 000 $\mathrm{kg(fuel)}$,
respectively, over those of a wind-optimal strategy (this strategy is regarded as an economically optimal strategy; see Sect. 2.4
of Yamashita et al., 2016). This study clearly indicated the fuel increase by avoiding contrail formations. Our results agree well
with the finding of Sridhar et al. (2013).

Table 4 clearly shows a trade-off between economic cost and climate impact (see also Supplement Fig. S4). Compared to
the COC option, the climate option decreases $ATR20_{total}$ by 67.9 % with an additional COC of 9.8 %. A similar trade-off
certainly exists between the minimum COC and the minimum climate impact trajectories for each airport pair. The trade-off
obtained from our results agrees with that indicated by many studies (see Sect. 1). Moreover, Niklaß et al. (2017) performed an
aircraft trajectory optimization for nine north-Atlantic flight routes varying weighting factors on average temperature response
over 100 years (ATR100) and on COC under ISA conditions. They showed a clear trade-off between the cost and the climate
impact. The minimum climate impact trajectories, on average, reduced ATR100 by 28.4 % with an additional COC of 7.1 %,
compared to those measures of the minimum COC trajectories. Our results agree with those shown by Niklaß et al. (2017). As
discussed above, the many previous studies corroborate the consistency of the AirTraf simulations.

**5  Conclusions**

We introduced updates to the air traffic simulation model AirTraf in the chemistry-climate model EMAC. The submodel AirTraf
2.0 was developed according to the MESSy standard and was described in detail in this paper. This submodel introduces seven
new aircraft routing options for air traffic simulations: the fuel use, the $NO_x$ emission, the $H_2O$ emission, the contrail formation,

the simple operating cost, the cash operating cost, and the climate impact options. Our flight trajectory optimization methodology consists of genetic algorithms; the methodology was similarly used and was validated beforehand (Yamashita et al., 2016). The particular strength of AirTraf is to enable a flight trajectory optimization for a global flight movement set in the atmosphere which is comprehensively described by EMAC. The novel routing option, i.e., the climate impact option, has been integrated in AirTraf 2.0. This option uses meteorological variables in terms of (spatially and temporally varying) aviation climate impact estimated by the aCCFs, and optimizes flight trajectories by minimizing their anticipated climate impact. As the aCCFs are new proxies for the climate-optimized routing, AirTraf takes a role in verifying the aCCFs themselves and the climate impact option based on the aCCFs in multi-annual (long-term) simulations.

To test the submodel AirTraf 2.0, example simulations were carried out with 103 north-Atlantic flights of an Airbus A330 aircraft for a typical winter day. AirTraf 2.0 simulates the one-day air traffic successfully for the newly developed routing option concerning different optimization objectives, e.g., contrail avoidance, cash operating cost, and climate impact (represented by average temperature response over 20 years), and finds the different families of optimum flight trajectories, which minimize the corresponding objective functions. The characteristics of these routing options include that aircraft is flown as the minimum economic cost with both, the SOC and the COC options. These options are comparably effective for economic cost indices. AirTraf 2.0 differentiates the minimum time, the minimum fuel, and the minimum COC options. The COC option lies between the minimum time and the minimum fuel options, and thus minimizes COC by taking the best compromise between the flight time and the fuel use into account. The $NO_x$ option minimizes $NO_x$ emission; this option differs from the fuel and the COC options. The contrail and the climate options decrease the climate impact (indicated by $ATR20_{total}$), which causes extra operating costs. A trade-off between the cost and the climate impact certainly exists. Compared to the COC option, the climate and the contrail options decrease $ATR20_{total}$ by 67.9 and by 43.4 % with an increase of COC by 9.8 and by 12.0 %, respectively. Thus, the climate option seems to be more effective on the cost-benefit performance than the contrail option. We believe that these climate benefits are most likely an upper limit. The simulation results were compared with literature data. The relative changes in the performance measures among the various routing options agree well in sign with those shown by many previous studies. This comparison has limitations because of different methodologies, different atmospheric conditions, etc. Nonetheless, the many literature data offer evidence to indicate the consistency of the AirTraf simulations.

The integration of AirTraf into EMAC allows one to optimize flight trajectories and to study aircraft routings under historical, present-day and future conditions of the climate system. We acknowledge that the simulation results depend on the atmospheric conditions of the target day. Thus, it is important to examine whether the findings, e.g., the trade-off between the cost and the climate impact, are common under any atmospheric conditions. Recently, Yamashita et al. (2020) examined this for representative weather types over the North Atlantic by using EMAC with AirTraf 2.0. Furthermore, the integrated aircraft routing options could be extended to conflicting scenarios. Yin et al. (2018a) investigated a trade-off between flight time and contrail formation for trans-Atlantic flights, by combining the time and the contrail options. Another option could easily be created by adding a corresponding objective function. The AirTraf development presented in this paper leads to a further detailed understanding of characteristics of various aircraft routing strategies.

## Appendix A:  The algorithmic Climate Change Functions

The aCCFs are calculated by the submodel ACCF (version 1.0). The derivation and validation of the aCCFs of ozone, methane, water vapour have been published by Van Manen (2017), Yin et al. (2018b), and Van Manen and Grewe (2019); the aCCF of contrails is described by Yin et al. (manuscript in preparation, 2020). The aCCFs for ozone, methane, water vapour, $CO_2$ and contrails are formulated as follows:

$$\text{aCCF}_{O_3} = \begin{cases} -5.20 \times 10^{-11} + 2.30 \times 10^{-13}T + 4.85 \times 10^{-16}\Phi - 2.04 \times 10^{-18}T\Phi, & \text{if aCCF}_{O_3} > 0, \\ 0, & \text{if aCCF}_{O_3} \leq 0, \end{cases} \tag{A1}$$

$$\text{aCCF}_{CH_4} = \begin{cases} -9.83 \times 10^{-13} + 1.99 \times 10^{-18}\Phi - 6.32 \times 10^{-16}F_{in} + 6.12 \times 10^{-21}\Phi F_{in}, & \text{if aCCF}_{CH_4} \leq 0, \\ 0, & \text{if aCCF}_{CH_4} > 0, \end{cases} \tag{A2}$$

$$\text{aCCF}_{H_2O} = 4.05 \times 10^{-16} + 1.48 \times 10^{-16}|PV|, \tag{A3}$$

$$\text{aCCF}_{CO_2} = 6.35 \times 10^{-15}, \tag{A4}$$

$$\text{aCCF}_{\text{contrail}} = \begin{cases} 1.0 \times 10^{-10}(0.0073 \times (10^{0.0107T} - 1.03)) \times 0.114, & \text{if Potcov} > 0 \text{ .and. nighttime}, \\ 1.0 \times 10^{-10}(-1.7 - 0.0088\text{OLR}) \times 0.114, & \text{if Potcov} > 0 \text{ .and. daytime}, \\ 0, & \text{if Potcov} \leq 0, \end{cases} \tag{A5}$$

where $T$ is the atmospheric temperature in K, $\Phi$ is the geopotential in $\text{m}^2\text{s}^{-2}$, $F_{in}$ is the incoming solar radiation at the top of atmosphere in $\text{Wm}^{-2}$, $PV$ is the potential vorticity in PVU ($1\,\text{PVU} = 10^{-6}\text{Km}^2\text{kg}^{-1}\text{s}^{-1}$), and OLR is the outgoing longwave radiation in $\text{Wm}^{-2}$. Given values of these meteorological variables, Eqs. (A1) and (A2) yield $\text{aCCF}_{O_3}$ and $\text{aCCF}_{CH_4}$ in $\text{K}(\text{kg}(NO_2))^{-1}$; Eqs. (A3) and (A4) yield $\text{aCCF}_{H_2O}$ and $\text{aCCF}_{CO_2}$ in $\text{K}(\text{kg}(\text{fuel}))^{-1}$; and Eq. (A5) yields $\text{aCCF}_{\text{contrail}}$ in $\text{K}(\text{km}(\text{contrail}))^{-1}$. The $\text{aCCF}_{CO_2}$ is the sole constant value (Dahlmann, 2018). The $\text{aCCF}_{CO_2}$ is calculated by using the non-linear climate-chemistry response model AirClim (Grewe and Stenke, 2008; Dahlmann, 2012; Dahlmann et al., 2016), assuming a 1 Tg fuel use in 2010 with the annual growth rate according to the future global aircraft scenario Fa1 (Penner et al., 1999). The $\text{aCCF}_{CO_2}$ is the averaged temperature response of $CO_2$ for the period $2010-2029$ (in K per kilogram of fuel) calculated by AirClim. The $\text{aCCF}_{\text{contrail}}$ for the night-time contrails takes positive values; if the temperature is less than 201 K, $\text{aCCF}_{\text{contrail}}$ for the night-time contrails is set to zero. The $\text{aCCF}_{\text{contrail}}$ for the day-time contrails can take positive and negative values, depending on the OLR (the threshold is $-193.18\,\text{Wm}^{-2}$). As for the time boundaries of day and night, the local time and solar zenith angle are calculated for locations where contrails could form ($\text{Potcov} > 0$). For locations in darkness, the time of sunrise is then calculated. If the time between the local time and sunrise is greater than six hours, the $\text{aCCF}_{\text{contrail}}$ for the night-time contrails is applied. If the contrail forms in daylight, or in darkness but with less than six hours before sunrise, the $\text{aCCF}_{\text{contrail}}$ for the day-time contrails is applied. These calculations are performed online in EMAC by the submodel

ACCF. In AirTraf 2.0, those five aCCFs are calculated as flight properties for waypoints and then the corresponding ATR20s are calculated for flight segments (see Table 2).

*Code and data availability.* AirTraf is implemented as a submodel of the Modular Earth Submodel System (MESSy). MESSy is continuously further developed and applied by a consortium of institutions. The usage of MESSy and access to the source code is licenced to all affiliates of institutions which are members of the MESSy Consortium. Institutions can become a member of the MESSy Consortium by signing the MESSy Memorandum of Understanding. More information can be found on the MESSy Consortium Website (http://www.messy-interface.org). The submodel AirTraf 2.0 presented here has been developed on the basis of MESSy version 2.53 and is available since the official release of MESSy version 2.54. The status information for AirTraf including the license conditions is available on the website. The data from the simulations will be provided by the authors on request.

*Author contributions.* HY, FY and VG designed the submodel AirTraf 2.0. HY, FY, PJ, SM, and BK implemented the coupling of AirTraf 2.0 with the Modular Earth Submodel System (MESSy). FY, VG, SM, KD, and CF developed the algorithmic Climate Change Functions (aCCFs). FY and VG designed the submodel ACCF. HY performed the simulations and analyzed the results presented in this paper.

*Competing interests.* The authors declare that they have no conflict of interest.

*Acknowledgements.* This study was supported by the project ATM4E (Air Traffic Management for Environment; https://www.atm4e.eu/) funded from the SESAR Joint Undertaking under grant agreement No. 699395 under European Union's Horizon 2020 research and innovation program. This study was also supported by the DLR project WeCare (Utilizing Weather Information for Climate Efficient and Eco Efficient Future Aviation) and Eco2Fly. The flight plan was provided by the European Union FP7 Project REACT4C (Reducing Emissions from Aviation by Changing Trajectories for the Benefit of Climate; https://www.react4c.eu/). We gratefully acknowledge the computational resources for the simulations, which were provided by the Deutsches Klimarechenzentrum (DKRZ). We wish to acknowledge the valuable contributions of the Department of Meteorology at the University of Reading, especially Emma Irvine, for her development of the $\mathrm{aCCF_{contrail}}$. We also wish to acknowledge our colleagues, especially Robert Sausen, for his support of the projects. We wish to thank Malte Niklaß for providing comparison data on the trajectory optimization. We wish to thank Duy Sinh Cai for his invaluable help on the model development. We would like to thank Anton Stephan for providing an internal review. We would like to express our gratitude to anonymous reviewers for their helpful comments and discussion.

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

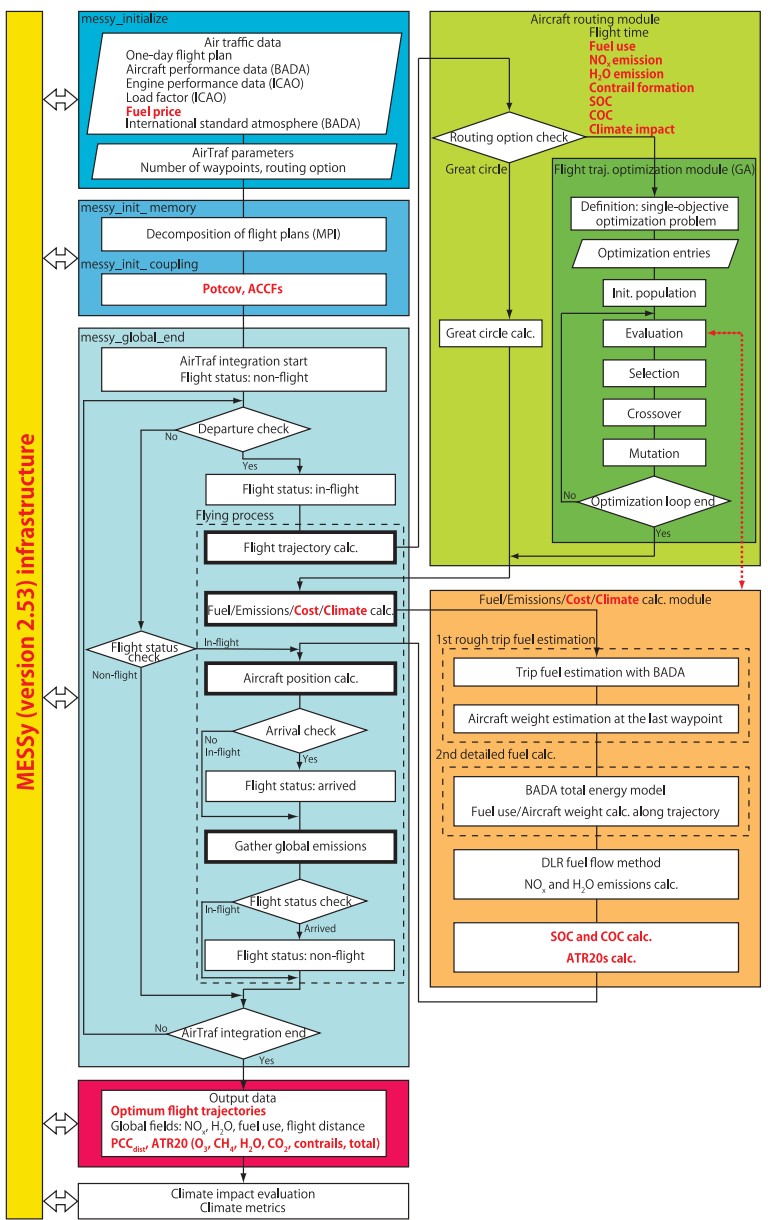

**Figure 1.** Updated flowchart of the MESSy submodel AirTraf 2.0 (updates from AirTraf 1.0 are highlighted by red texts and arrows). MESSy as part of EMAC provides interfaces (yellow) to couple various submodels for data exchange, run control and data input/output. AirTraf 2.0 is coupled to the submodel CONTRAIL (version 1.0; Frömming et al., 2014) and the submodel ACCF (version 1.0). Air traffic data and AirTraf parameters are imported in the initialization phase (`messy_initialize`, dark blue). AirTraf includes the flying process in `messy_global_end` (dashed box, light blue), which comprises four main computation procedures (bold-black boxes). AirTraf uses three modules: the aircraft routing module (light green), the fuel-emissions-cost-climate calculation module (light orange), and the flight trajectory optimization module (dark green). Resulting optimum flight trajectories and global fields of flight properties are output (rose red).

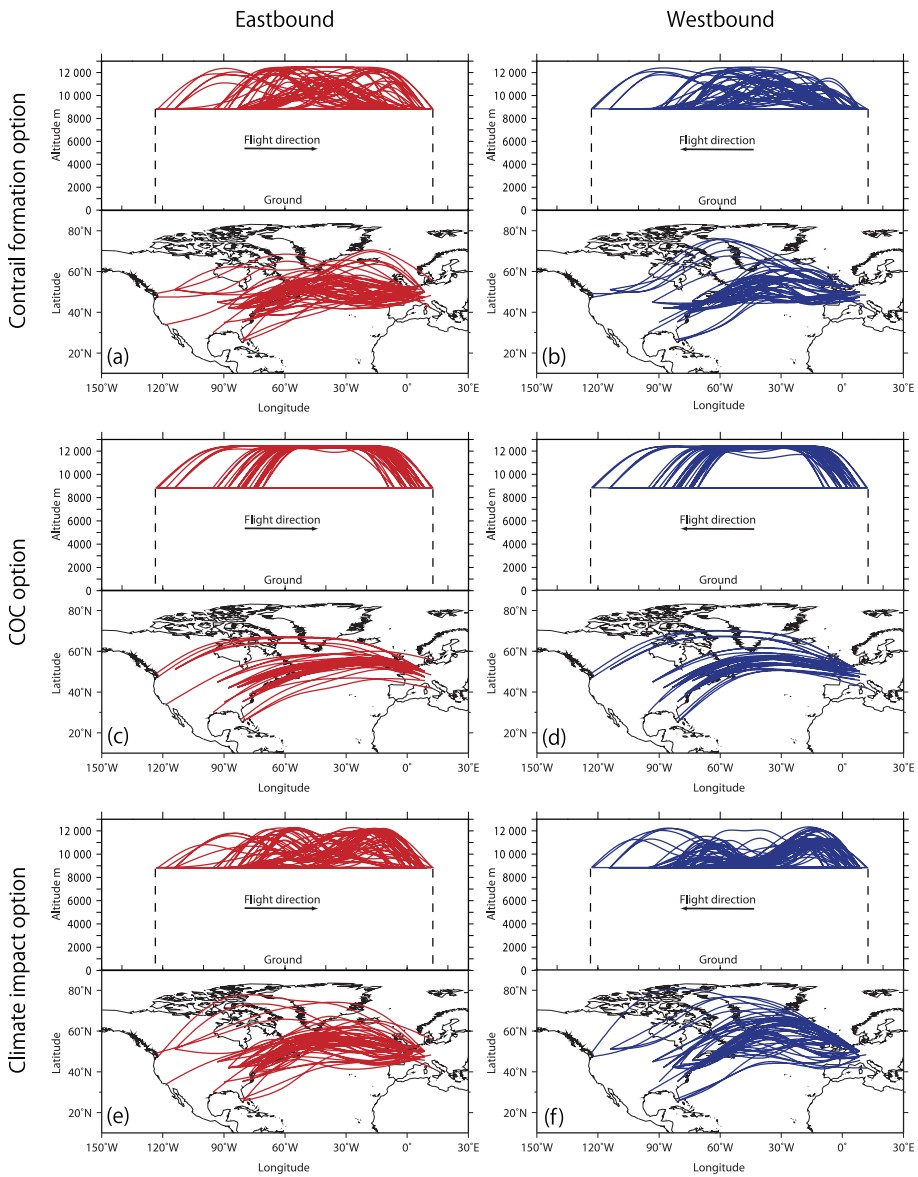

**Figure 2.** Optimized flight trajectories from a one-day AirTraf simulation (52 eastbound and 51 westbound flights) for the contrail formation (a, b), the COC (c, d), and the climate impact routing options (e, f). For each figure, the trajectories are shown in the vertical cross-section (top) and projected on the ground (bottom).

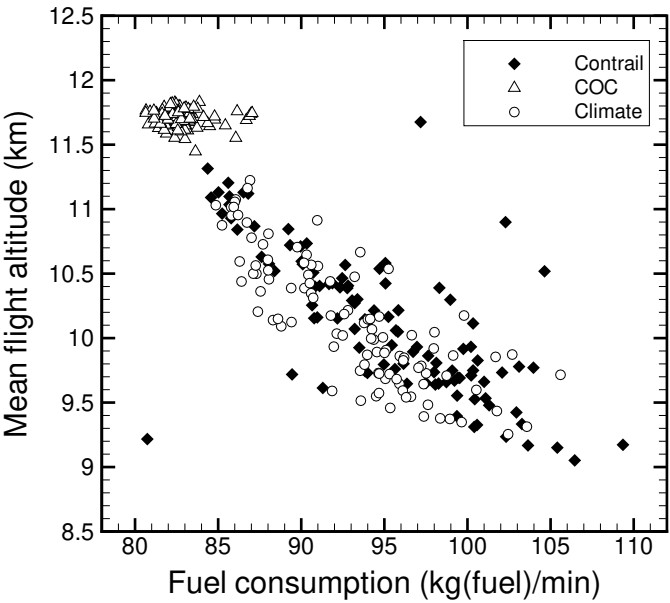

**Figure 3.** Mean fuel consumption vs. mean flight altitude for 103 individual flights obtained by the contrail formation, the COC and the climate impact routing options.

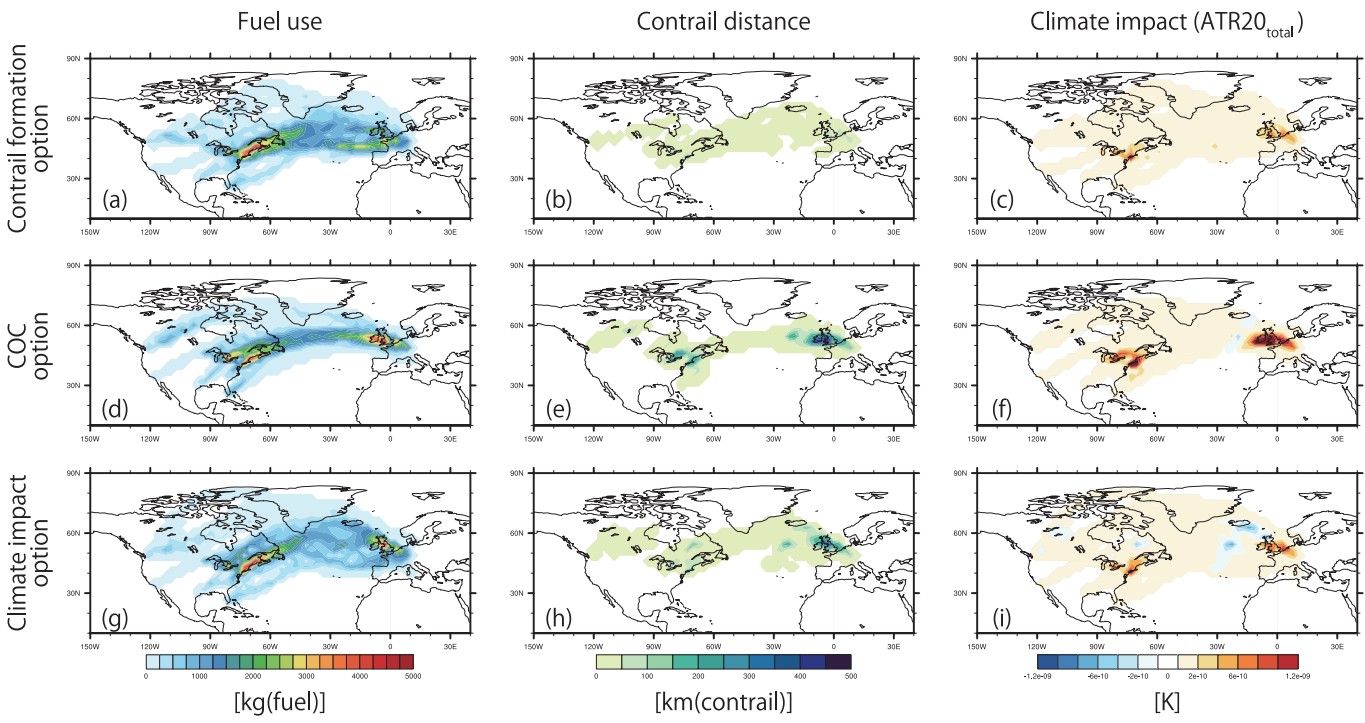

**Figure 4.** Vertically integrated distribution of fuel use, contrail distance, and climate impact indicated by $ATR20_{total}$ during the day (from December 1, 2015 00:00:00 to December 2, 2015 00:00:00 UTC). Top: contrail formation option. Middle: COC option. Bottom: climate impact option. These distributions were obtained with the optimized flight trajectories shown in Fig. 2 (sum of 103 flights).

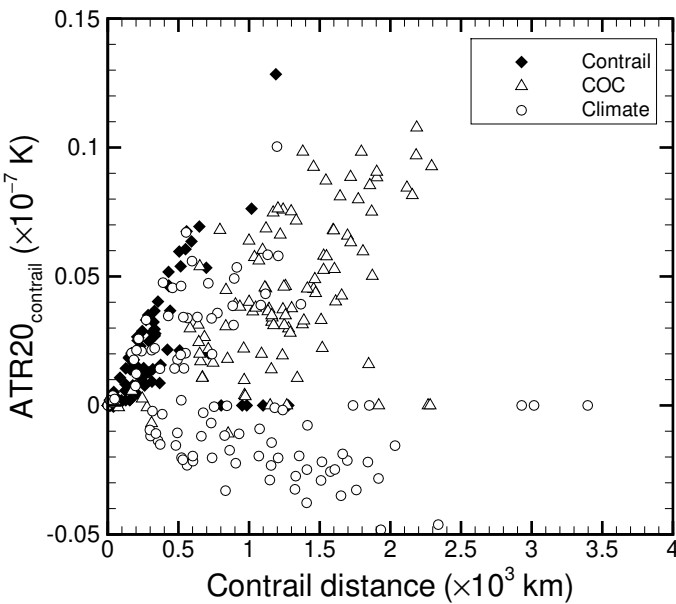

**Figure 5.** Contrail distance vs. ATR20$_{\text{contrail}}$ for 103 individual flights obtained by the contrail formation, the COC and the climate impact routing options.

**Table 1.** Relevant data of an Airbus A330-301 aircraft and constant parameters applied for AirTraf 2.0. The column "New in V2.0" denotes parameters newly introduced in AirTraf 2.0.

| Parameter | Value | Unit | New in V2.0 | Description |
|---|---|---|---|---|
| AFW | 103 070 | kg | x | Airframe weight estimated by $\text{AFW} = \text{MEW} - N_\text{eng}\text{EDW}$ |
| $c_t$ | 0.75 | (USDollar)s$^{-1}$ | x | Unit time costs[a] |
| $c_f$ | 0.51 | (USDollar)kg$^{-1}$ | x | Unit fuel costs[a] |
| $C_\text{D0}$ | 0.019805 | – | | Parasitic drag coef. (cruise)[b] |
| $C_\text{D2}$ | 0.031875 | – | | Induced drag coef. (cruise)[b] |
| $C_\text{f1}$ | 0.61503 | kg min$^{-1}$kN$^{-1}$ | | First thrust specific fuel consumption (TSFC) coef. (jet engines)[b] |
| $C_\text{f2}$ | 919.03 | kt | | Second TSFC coef.[b] |
| $C_\text{fcr}$ | 0.93655 | – | | Cruise fuel flow correction coef.[b] |
| EDW | 5091.62 | kg | x | Engine dry weight. CF6-80E1A2 engine[c] |
| $\text{EINO}_{x,\text{ref}}$ | 4.88; 12.66; 22.01; 28.72 | g($\text{NO}_x$)(kg(fuel))$^{-1}$ | | Reference $\text{NO}_x$ emission index at take off, climb out, approach and idle conditions (sea level). CF6-80E1A2 (2GE051)[d] |
| $\text{EIH}_2\text{O}$ | 1230 | g($\text{H}_2\text{O}$)(kg(fuel))$^{-1}$ | | $\text{H}_2\text{O}$ emission index[e] |
| $f_\text{ref}$ | 0.228; 0.724; 2.245; 2.767 | kg(fuel)s$^{-1}$ | | Reference fuel flow at take off, climb out, approach and idle conditions (sea level). CF6-80E1A2 (2GE051)[d] |
| $g$ | 9.8 | ms$^{-2}$ | | Gravity acceleration |
| JFD | 0.804 | kgl$^{-1}$ | x | Jet fuel density at 15°C (Jet A-1) |
| JFP | 0.41 | (USDollar)l$^{-1}$ | x | Jet fuel price[f] |
| $M$ | 0.82 | – | | Cruise Mach number[b] |
| MEW | 113 253 | kg | x | Baseline manufactures empty weight. $\text{MEW} = 0.9053\text{OEW}$[g] |
| MPL | 47 900 | kg | | Maximum payload[b] |
| MTOGW | 212 000 | kg | x | Maximum take-off weight[h] |
| $N_\text{seat}$ | 295 | – | x | Number of seats (3-class)[i] |
| $N_\text{eng}$ | 2 | – | x | Number of engines[h] |
| OEW | 125 100 | kg | | Operational empty weight[b] |
| OLF | 0.62 | – | | ICAO overall (passenger/freight/mail) weight load factor in 2008[j] |
| $P_0$ | 101 325 | Pa | | Reference pressure (sea level) |
| $r_\text{inf}$ | 2.28 | % | x | Ave. U.S. inflation rate (1994-2014)[k] |
| $R$ | 287.05 | JK$^{-1}$kg$^{-1}$ | | Gas constant for dry air |
| $S$ | 361.6 | m$^2$ | | Reference wing surface area[b] |
| SLST | 268.7 | kN | x | Thrust per engine (maximum continuous). CF6-80E1A2[h] |
| SPD | 86 400 | sday$^{-1}$ | | Time (Julian date) $\times$ SPD = Time (s) |
| $T_0$ | 288.15 | K | | Reference temperature (sea level) |
| $Y_\text{pre}$ | 2015 | year | x | Present year for COC calculation |
| $Y_\text{ref}$ | 1993 | year | x | Reference year for COC calculation |
| $\gamma$ | 1.4 | – | | Adiabatic gas constant |

[a] Michael A. (2015); [b] Eurocontrol (2011); [c] EASA (2011); [d] ICAO (2005); [e] Penner et al. (1999); [f] IATA (2017); [g] MEW was estimated, because the exact value was unavailable; [h] EASA (2013); [i] Aircraft Commerce (2008); [j] Anthony (2009); [k] IMF (2016)

**Table 2.** Properties assigned to a resulting flight trajectory. The properties of the three groups (divided by rows) are obtained from the nearest grid box of EMAC at departure time of the flight, the flight trajectory calculation (Fig. 1), and the fuel-emissions-cost-climate calculation (Fig. 1; some properties are calculated in flight trajectory optimizations depending on a selected routing option), respectively. The attribute type indicates where the values of properties are allocated. "W", "S" and "T" stand for waypoints ($i = 1, 2, \cdots, n_{\mathrm{wp}}$), flight segments ($i = 1, 2, \cdots, n_{\mathrm{wp}} - 1$), and a whole flight trajectory in column 3, respectively. The column "New in V2.0" denotes properties newly introduced in AirTraf 2.0.

| Property | Unit | Attribute type | New in V2.0 | Description |
|---|---|---|---|---|
| $\mathrm{aCCF_{O_3}}$ | $\mathrm{K(kg(NO_2))^{-1}}$ | W | x | Algorithmic Climate Change Function of ozone[a][b]. See Eq. (A1) |
| $\mathrm{aCCF_{CH_4}}$ | $\mathrm{K(kg(NO_2))^{-1}}$ | W | x | Algorithmic Climate Change Function of methane[a][b]. See Eq. (A2) |
| $\mathrm{aCCF_{H_2O}}$ | $\mathrm{K(kg(fuel))^{-1}}$ | W | x | Algorithmic Climate Change Function of water vapor[a][b]. See Eq. (A3) |
| $\mathrm{aCCF_{CO_2}}$ | $\mathrm{K(kg(fuel))^{-1}}$ | W | x | Algorithmic Climate Change Function of $\mathrm{CO_2}$[c]. See Eq. (A4) |
| $\mathrm{aCCF_{contrail}}$ | $\mathrm{K(km(contrail))^{-1}}$ | W | x | Algorithmic Climate Change Function of contrails[d]. See Eq. (A5) |
| Potcov | fraction | W | x | Potential persistent contrail cirrus coverage[e] |
| $P$ | Pa | W | | Pressure |
| $T$ | K | W | | Temperature |
| $\rho$ | $\mathrm{kg\,m^{-3}}$ | W | | Air density |
| $u, v, w$ | $\mathrm{m\,s^{-1}}$ | W | | Three dimensional wind components |
| $a$ | $\mathrm{m\,s^{-1}}$ | W | | Speed of sound |
| $\mathrm{ATR20_{O_3}}$ | K | S | x | Anticipated climate impact of ozone. See Eq. (8) |
| $\mathrm{ATR20_{CH_4}}$ | K | S | x | Anticipated climate impact of methane. See Eq. (9) |
| $\mathrm{ATR20_{H_2O}}$ | K | S | x | Anticipated climate impact of water vapor. See Eq. (10) |
| $\mathrm{ATR20_{CO_2}}$ | K | S | x | Anticipated climate impact of $\mathrm{CO_2}$. See Eq. (11) |
| $\mathrm{ATR20_{contrail}}$ | K | S | x | Anticipated climate impact of contrails. See Eq. (12) |
| $\mathrm{ATR20_{total}}$ | K | S | x | Anticipated climate impact (total). See Eq. (13) |
| $d$ | m | S | | Flight distance |
| ETO | Julian date | W | | Estimated time over |
| FT | s | T | | Flight time. $\mathrm{FT} = (\mathrm{ETO}_{n_{\mathrm{wp}}} - \mathrm{ETO}_1) \times \mathrm{SPD}$ |
| $h$ | m | W | | Flight altitude |
| $\overline{h}$ | m | T | | Mean flight altitude. $\overline{h} = 1/n_{\mathrm{wp}} \sum_{i=1}^{n_{\mathrm{wp}}} h_i$ with waypoint number $n_{\mathrm{wp}}$. |
| $\mathrm{PCC_{dist}}$ | km(contrail) | S | x | Contrail distance[f] |
| $V_{\mathrm{TAS}}$ | $\mathrm{m\,s^{-1}}$ | W | | True air speed |
| $V_{\mathrm{ground}}$ | $\mathrm{m\,s^{-1}}$ | W | | Ground speed |
| $\lambda$ | deg | W | | Longitude |
| $\phi$ | deg | W | | Latitude |
| COC | USDollar | T | x | Cash operating cost[g] |
| $\mathrm{EINO}_{x,a}$ | $\mathrm{g(NO_x)(kg(fuel))^{-1}}$ | W | | $\mathrm{NO}_x$ emission index |
| $F_{\mathrm{cr}}$ | $\mathrm{kg(fuel)\,s^{-1}}$ | W | | Fuel flow of an aircraft (cruise) |
| FUEL | kg | S | | Fuel use |
| $\mathrm{H_2O}$ | $\mathrm{g(H_2O)}$ | S | | $\mathrm{H_2O}$ emission |
| $m$ | kg | W | | Aircraft weight |
| $\mathrm{NO}_x$ | $\mathrm{g(NO_x)}$ | S | | $\mathrm{NO}_x$ emission |
| SOC | USDollar | T | x | Simple operating cost |

[a] Van Manen (2017); [b] Van Manen and Grewe (2019); [c] Dahlmann (2018); [d] Yin et al. (manuscript in preparation, 2020); [e] Frömming et al. (2014); [f] Yin et al. (2018a); [g] Liebeck et al. (1995)

**Table 3.** Setup for AirTraf one-day simulations. The setups of the two groups (divided by rows) are used for AirTraf/EMAC and for AR-MOGA (Sasaki et al., 2002; Sasaki and Obayashi, 2004, 2005), respectively. $\alpha$ is an user-specified crossover parameter; $r_m$ is a mutation rate; and $\eta_m$ is an parameter controlling the shape of a probability distribution. Details of these parameters are described in Yamashita et al. (2016).

| Parameter | Description |
|---|---|
| ECHAM5 resolution | T42L31ECMWF (2.8° by 2.8°) |
| Simulation period | December 1, 2015 00:00:00 − December 2, 2015 00:00:00 UTC |
| Time step of EMAC | 12 min |
| Flight plan | 103 trans-Atlantic flights (eastbound 52/westbound 51)[a] |
| Aircraft type | A330-301 |
| Engine type | CF6-80E1A2, 2GE051 (with 1862M39 combustor) |
| Flight altitude changes | [FL290, FL410] (fixed at FL350 for the great circle option) |
| Mach number | 0.82 |
| Number of waypoints, $n_{wp}$ | 101 |
| Design variable, $n_{dv}$ | 11 (6 locations and 5 altitudes) |
| Population size, $n_p$ | 100 |
| Number of generations, $n_g$ | 100 |
| Selection | Stochastic universal sampling |
| Crossover | Blend crossover BLX-0.2 ($\alpha = 0.2$) |
| Mutation | Revised polynomial mutation ($r_m = 0.1$; $\eta_m = 5.0$) |

$a$ Grewe et al. (2014a) and REACT4C (2014)

**Table 4.** The nine performance measures obtained from the one-day AirTraf simulations with different aircraft routing options (the values indicate the sum of 103 flights). The minimum values of each performance measure are marked with an asterisk; changes (in %) relative to the COC option are given in parentheses. Bar charts of the same data are given in Fig. S4 in the Supplementary material.

| Routing option | Flight distance $10^3$km | Flight time h | Fuel use ton | $NO_x$ emission ton | $H_2O$ emission ton | Contrail distance $10^3$km | SOC Mil.USD | COC Mil.USD | ATR20$_{total}$ $10^{-7}$K |
|---|---|---|---|---|---|---|---|---|---|
| Great circle | 660.3* (−0.4) | 757.4 (+0.1) | 3979.1 (+5.8) | 44.6 (+5.5) | 4894.2 (+5.8) | 154.9 (+19.1) | 4.072 (+2.9) | 5.463 (+2.1) | 6.85 (+12.5) |
| Flight time | 663.2 (+0.02) | 739.4* (−2.3) | 4521.9 (+20.2) | 57.8 (+36.8) | 5562.0 (+20.2) | 127.7 (−1.9) | 4.299 (+8.6) | 5.673 (+6.1) | 10.44 (+71.5) |
| Fuel use | 663.3 (+0.03) | 757.3 (+0.1) | 3758.5* (−0.1) | 42.2 (−0.2) | 4623.0 (−0.1) | 128.5 (−1.2) | 3.960 (+0.03) | 5.351 (+0.03) | 5.85 (−3.9) |
| $NO_x$ emission | 664.5 (+0.2) | 758.8 (+0.3) | 3766.8 (+0.1) | 42.1* (−0.5) | 4633.1 (+0.1) | 131.8 (+1.3) | 3.968 (+0.2) | 5.360 (+0.2) | 5.83 (−4.2) |
| $H_2O$ emission | 663.3 (+0.03) | 757.3 (+0.1) | 3758.5 (−0.1) | 42.2 (−0.2) | 4623.0* (−0.1) | 128.5 (−1.2) | 3.960 (+0.03) | 5.351 (+0.03) | 5.85 (−3.9) |
| Contrail formation | 717.4 (+8.2) | 812.3 (+7.4) | 4625.5 (+23.0) | 57.0 (+34.9) | 5689.3 (+23.0) | 26.3* (−79.8) | 4.549 (+14.9) | 5.990 (+12.0) | 3.45 (−43.4) |
| SOC | 663.2 (+0.02) | 756.6 (+0.03) | 3760.4 (−0.02) | 42.2 (−0.1) | 4625.3 (−0.02) | 130.2 (+0.1) | 3.959* (0.0) | 5.349 (0.0) | 6.02 (−1.1) |
| COC | 663.1 | 756.4 | 3761.1 | 42.3 | 4626.2 | 130.1 | 3.959 | 5.349* | 6.09 |
| Climate impact | 703.2 (+6.0) | 801.4 (+5.9) | 4474.0 (+19.0) | 52.3 (+23.8) | 5503.1 (+19.0) | 92.6 (−28.8) | 4.443 (+12.2) | 5.874 (+9.8) | 1.96* (−67.9) |