# Peer review of "Newly developed aircraft routing options for air traffic simulation in the chemistry-climate model EMAC 2.53: AirTraf 2.0"

_Geoscientific Model Development, 2019_

## Referee Comment (RC1) · Anonymous Referee #1 · 3 Feb 2020

This paper documents the AirTraf version 2 submodel of the EMAC chemistry-climate model, developed to enable simulation of global air traffic in a climate model in order to investigate optimized routing strategies for the aviation sector. A set of one day simulations are run, showing that the model gives plausible output and the results are discussed in the context of previous literature. While the topic of abatement strategies for reducing aviation's climate impact is both important and current, and this modeling framework is a useful tool in this regard, the paper is not of a sufficient quality for publication in its current form. In general, the main messages can be polished and highlighted better. The introduction is long and unstructured, and it's difficult to extract the essence of what's new in this work (and why it's important). This does not really

get much clearer in the methods where most of AirTraf2 seems to follow AirTraf1 and is mostly described in Yamashita et al. 2016. While the discussion section is quite good, the results is only one page out of a 14-page paper, which is not quite convincing. The paper also needs substantial additional work to improve the writing and language. There are number strange formulations, short sentences and imprecise use of terminology that make the paper difficult to follow at times. Some examples are given below, but a general language check/ copyediting is recommended.

Selected specific comments: Title: suggest removing "Various". Makes it seem vague.

Abstract: Line 1: Add "the" before "climate impact of aviation (…)" Line 6-9: unclear, I don't really understand what the important result here is

Pg1: Line 16: The sentence starting with "the aviation sector is not" is redundant as you've just said that aviation contributes only 5% total climate impact. Line 23: a more up-to-date reference would be the Brasseur et al. 2016 paper in BAMS.

Pg2: Line 1: I don't understand the rationale behind introducing the terminology radiative impact (RI) instead of keeping well-established radiative forcing (RF). This is confusing and adds nothing to the paper. Please explain or change. Line 5: there are number of more recent studies showing higher contrail-cirrus forcing, reflecting more recent emission inventories. One example is the 2016 paper by Bock and Burkhardt in JGR-A. Such work should be reflected. Line 6: "Here the difference between time scales (…)": suggest removing, no point in telling the reader what you will tell them next. Line 6: "The emitted CO2 (…)" – this is not precise; the emitted CO2 does not have century-long timescale, the perturbation does. Line 7: "the impact is proportional to (…)": this may be true for emission, and perhaps even for RF, but when approximating fuel with temperature impact or other climate change seems doubtful. Line 10: the recent work by Lund et al. 2017 ESD include all components and show how this translates into temperature impacts. Could be a useful references. Line 17: Why is climate-optimized routing limited to the present-day fleet? Line 22: because of the

long residence time of CO2, its impact is the same regardless of location of emission. Please be more precise. Line 22: Please add a more detailed definition of CCS as the reader needs this later on. Line 24: Another strange sentence to suddenly introduce here instead of adding above when listing aviation non-CO2 effects. Line 29: what about trade-offs between e.g., contrail avoidance and increased fuel use?

Pg.3 Line 2: Presumably this is global-mean temperature response? Please specify. Line 5: what about the other way around, does a cost-optimized route increase climate impact? Line 6: do you mean using different emission metrics, of which AGTP is one? And which other metrics do you find in the literature? Here you only describe one approach. (from here on I do not list language issues, but note that there are a number of them also in the next pages...)

Pg. 8: Section 2.5.4: The treatment of contrail-cirrus is quite essential for routing strategies and I would like to see some more details of how this is done and what the limitations are (e.g., natural cloud suppression, life cycle etc.) here, not just a reference to earlier work.

Pg. 9 Line 20: ATR20 needs a definition. Is it calculated based on input of RF? What is assumed for contrail-cirrus properties? Line 26: But ATR20 is an average over 20 years? How can values be negative when the overall contrail-cirrus effect is a warming? Perhaps related to the above comment. . .

Pg.10: Line 3: how sensitive are results and conclusions to the running of only one day? E.g., dependence on meteorological conditions that day? Line 11: showing direct results is not a verification of simulations output. Line 21: over what time frame is the km coverage estimated? Integrated over the 1-day simulations?

Pg.11: Line 2-3: this is a very strange argument for correctness

Pg. 14: Line 10-11: how well does the treatment of contrails work for longer time integrations (in particular decades as mentioned earlier)? Is the potcov based on presentday conditions? Line 5-10: this type of information would be useful in the introduction.

---

## Referee Comment (RC2) · Anonymous Referee #2 · 3 Mar 2020

This paper proposes an updated sub-model in the ECHAM Atmospheric Chemistry model for flight trajectory optimisation and is within the scope of the journal (GMD). The algorithm now enables the flight trajectory to be optimised based on various scenarios, which can assist relevant stakeholders and policymakers to evaluate the trade-off between economic costs and the overall climate impact. Such a tool is expected to become increasingly important as the focus shifts to minimising aviation's overall environmental impact, including both $CO_2$ and non-$CO_2$ emissions.

While the work is well structured and written, there are several major aspects in the model that were not adequately addressed and must be significantly improved. Therefore, I believe that major revisions are necessary before this paper is accepted for publication.

Major Comments

1) [Page 6, Line 12] What is the rationale for selecting a 20-year time horizon for the average temperature response (ATR20)? Given that a proportion of $CO_2$ can remain in the atmosphere for over a millennium [Ref.1], the ATR20 can lead to a large underestimation in the $CO_2$ climate impacts. To overcome this, it is suggested that the authors perform a sensitivity analysis on the reported results by considering the use of different ATR time horizons (i.e. 100 years and 1000 years).

2) [Page 8, Section 2.5.4 and Appendix A] While the methodology selected to model the contrail climate impact was commonly used in previous studies, its limitations should be acknowledged and discussed in the paper. An optimisation algorithm based on the contrail length might be overly simplistic because it does not account for differences in the contrail radiative forcing, lifetime and coverage area:

Firstly, Eq. (A5) assumes that contrails always cool during the day because it has a negative aCCFcontrail and ATR20contrail. However, this is not true as many other studies have shown that contrails can either warm or cool during the day, depending on meteorology (such as ambient cirrus), radiation, and the solar zenith angle.

Secondly, some contrails formed during the day could also have lifetimes of up to 19 hours [Ref.2] and persist through the night, subsequently turning to a warming contrail, but the methodology does not appear to have considered the contrail lifetime. In Eq. (A5), it is also unclear on the conditions/time boundaries which constitutes as day-time and night-time.

Thirdly, the ATR20contrail could also be influenced by contrail spreading and its coverage area. However, Eq. (A5) and "ATR20contrail = aCCFcontrail x PCCdist" does not account for the change in contrail coverage area. Further clarification on these aspects

are required.

3) [Page 10, Lines 17 to 19] The results and Figure 2 show that flight trajectories based on the cash operating cost (COC) optimisation and minimum fuel consumption always selects a higher cruising altitude. However, this is very likely due to limitations of BADA 3. According to Nuic et al.[Ref.3], BADA 3 has a tendency to underestimate aircraft fuel consumption at higher altitudes and Mach numbers as the compressibility effect and wave drag are not modelled. While I understand that a more accurate version of BADA (BADA 4) is available, obtaining access to it can be challenging. Despite this, the authors should include more discussion on the effects of BADA 3 on their results, as well as acknowledge the limitations of BADA 3.

4) [Page 11, Lines 9 to 11] "The contrail option shows the minimum contrail distance and decreases ATR20total.... This option allows aircraft to widely detour the potential contrail regions". This sentence requires further clarification: given that the authors mentioned in Page 9 Line 26 that the "ATR20contrail can take positive and negative values, because of the day-time and night-time contrail effects", it should be made clear in the discussion on if the algorithm: (i) actively forms cooling contrails during the day and avoids forming warming contrails during the night; or (ii) minimises the overall contrail length at all times.

Minor Comments

1) [Page 1, Line 22] Replace "non-volatile black carbon (BC or soot)" with "non-volatile particulate matter such as BC" for correctness in terminology [Ref.4]. This is because black carbon (BC) is a subset of non-volatile particulate matter (nvPM), while the term "soot" includes both nvPM (BC and metallic compounds) and organic compounds.

2) [Page 2, Line 7] The sentence, "The emitted CO2 has a long residence time (a century)", should be corrected. According to Joos et al.1, however, the emitted CO2 can remain in the atmosphere after a millennium.

3) [Page 3, Line 15] Remove "the" from this sentence "the today's aircraft routing focuses on the minimum economic cost".

4) [Page 8, Line 22] There appears to be inconsistencies in the acronyms: Ctime and Cfuel was used in line 22. However, ct and cf are used in Eq. (6). This can confuse future readers.

5) [Section 2.5.7] Please acknowledge the large uncertainties in the global temperature response, especially from contrails (ATR20contrail) due to uncertainties in the contrail efficacy[Ref.5,6].

6) [Section 4: Discussion] The authors should highlight that these results (climate benefits) is likely an upper limit, because airspace congestion and air traffic management could minimise the flexibility for flights to perform these trajectory optimisations.

7) [Eq. (5) and Table 1] Consider using different notations for the mass fuel flow rate (fref), as this is similar to the objective function (f) and can lead to confusion.

8) [Appendix A, Eq. (A5)] aCCFcontrail = . . ., if Potcov $\geq$ 0: should this be > 0 instead? Similarly, aCCFcontrail = 0, if Potcov < 0: should this be $\leq$ 0 instead?

References

1. Joos F, Roth R, Fuglestvedt JS, Peters GP, Enting IG, von Bloh W, Brovkin V, Burke EJ, Eby M, Edwards NR, Friedrich T, Frölicher TL, Halloran PR, Holden PB, Jones C, Kleinen T, Mackenzie FT, Matsumoto K, Meinshausen M, et al. Carbon dioxide and climate impulse response functions for the computation of greenhouse gas metrics: a multi-model analysis. Atmos Chem Phys. 2013;13(5):2793-2825. doi:10.5194/acp-13-2793-2013

2. Haywood JM, Allan RP, Bornemann J, Forster PM, Francis PN, Milton S, Rädel G, Rap A, Shine KP, Thorpe R. A case study of the radiative forcing of persistent contrails evolving into contrail‐induced cirrus. J Geophys Res Atmos. 2009;114(D24201). doi:doi:10.1029/2009JD012650

3. Nuic A, Poles D, Mouillet V. BADA: An advanced aircraft performance model for present and future ATM systems. Int J Adapt Control Signal Process. 2010;24(10):850-866. doi:10.1002/acs.1176

4. Petzold A, Ogren JA, Fiebig M, Laj P, Li S-M, Baltensperger U, Holzer-Popp T, Kinne S, Pappalardo G, Sugimoto N. Recommendations for reporting" black carbon" measurements. Atmos Chem Phys. 2013;13(16):8365-8379.

5. Hansen J, Sato M, Ruedy R, Nazarenko L, Lacis A, Schmidt GA, Russell G, Aleinov I, Bauer M, Bauer S, Bell N, Cairns B, Canuto V, Chandler M, Cheng Y, Genio A Del, Faluvegi G, Fleming E, Friend A, et al. Efficacy of climate forcings. J Geophys Res. 2005;110(D18):D18104. doi:10.1029/2005JD005776

6. Ponater M, Marquart S, Sausen R, Schumann U. On contrail climate sensitivity. Geophys Res Lett. 2005;32(10).

---

## Author Comment (AC1) · 10 Jun 2020

We are grateful to the referee #1 for the very helpful and encouraging comments on the original version of our manuscript. We took all comments into account and rewrote the manuscript accordingly. Here are our replies:

- **General comment:** This paper documents the AirTraf version 2 submodel of the EMAC chemistry-climate model, developed to enable simulation of global air traffic in a climate model in order to investigate optimized routing strategies for the aviation sector. A set of one day simulations are run, showing that the model gives plausible output and the results are discussed in the context of previous literature. While the topic of abatement strategies for reducing aviation's climate impact is both important and current, and this modeling framework is a useful tool in this regard, the paper is not of a sufficient quality for publication in its current form. In general, the main messages can be polished and highlighted better. The introduction is long and unstructured, and it's difficult to extract the essence of what's new in this work (and why it's important). This does not really get much clearer in the methods where most of AirTraf 2 seems to follow AirTraf1 and is mostly described in Yamashita et al. 2016. While the discussion section is quite good, the results is only one page out of a 14-page paper, which is not quite convincing. The paper also needs substantial additional work to improve the writing and language. There are number strange formulations, short sentences and imprecise use of terminology that make the paper difficult to follow at times. Some examples are given below, but a general language check/copyediting is recommended.

  Reply: We thank the referee #1 for the useful comments. We have addressed all the comments and structured our reply according to the reviewer's general comments into
  - a) Highlight improvements
  - b) Shortening and improved structure of the introduction
  - c) Methods: clarifying the improvements of AirTraf 2.0 over AirTraf 1.0
  - d) Extension of the results section
  - e) Language improvements
  - f) Modification of short sentences
  - g) Explanation of terminologies
  - h) Formula improvements
  - i) Modification of references.

We believe that this revision represents a polishing of the whole paper.

**a) Highlight improvements:**
To highlight the main messages of this paper, we rewrote the abstract and the conclusions. We also modified the introduction to improve the structure, and to show what's new in this work and why AirTraf 2.0 is important. Details are described in the "b) Shortening and improved structure of the introduction" below.

[Abstract]
**Aviation contributes to climate change and the** climate impact of aviation is expected to increase further. **Adaptions of** aircraft routings **in order to reduce the climate impact** are an important **climate change mitigation** measure .  **The air traffic simulator AirTraf, as a submodel of the ECHAM/MESSy Atmospheric Chemistry (EMAC) model, enables the evaluation of such measures. For the first version of the submodel AirTraf, we concentrated on the general set-up of the model, including departure and arrival, performance and emissions, and technical aspects such as the parallelization of the aircraft trajectory calculation with only a limited set of optimization possibilities (time and distance).**  **Here, in the second version of AirTraf, we focus on enlarging the objective functions by seven new options to enable assessing operational improvements in many more aspects including economic costs, contrail occurrence and climate impact. We verify that the AirTraf set-up, e.g. in terms of number and choice of design variables for the genetic algorithm, allows finding solutions even with highly structured fields such as contrail occurrence. This is shown by** example

simulations of the new routing options , **including** around 100 north-Atlantic flights of an Airbus A330 aircraft for a typical winter day. The results clearly show that **AirTraf 2.0 can find** the **different** famil**ies** of optimum flight trajectories (three-dimensional)  **for specific routing options; those trajectories minimize the corresponding objective functions successfully.**  The minimum cost option  **lies** between the minimum time and the minimum fuel  **options. Thus, aircraft operating costs are minimized by taking the best compromise between flight time and fuel use.** The aircraft routings for contrail avoidance and minimum climate impact reduce the potential climate impact, which is estimated by using algorithmic Climate Change Functions, whereas these two routings increase **the**  **aircraft** operating costs. A trade-off between the aircraft operating costs and the climate impact is confirmed. The simulation results are compared with literature data and the consistency of the submodel AirTraf 2.0 is verified.

[Conclusions]
We revised the conclusions to highlight the outcomes in a better way, e.g., on page 14 lines 25-32, "AirTraf 2.0 simulates the one-day air traffic successfully for **the newly developed routing option concerning different optimization objectives, e.g., contrail avoidance, cash operating cost, and climate impact (represented by average temperature response over 20 years), and finds the different families of optimum flight trajectories, which minimize the corresponding objective functions**. The characteristics of the**se** routing options  **include that A**ircraft **i**s flown as the minimum economic cost with both, the SOC and the COC options. These options are comparably effective for economic cost indices. The AirTraf 2.0 differentiates the minimum time, the minimum fuel, and the minimum COC  **options;.**  **t**The COC option  **lies** between the minimum time and the minimum fuel  **options., and thus minimizes COC by taking the best compromise between the flight time and the fuel use into account.** The $NO_x$ option minimizes $NO_x$ emission; this option differs from the fuel and the COC options. The contrail and the climate options decrease **the** climate impact (indicated by $ATR20_{total}$), which causes extra operating costs. A trade-off between the cost and the climate impact **certainly** exists."

**b) Shortening and improved structure of the introduction:**
To shorten the introduction and to improve its structure, we modified the text and structured the introduction into
- Background: the climate impact of aviation
- Introduction of a climate-optimized routing
- Previous studies: benefits of the climate-optimized routing
- Ultimate aims and introduction of the AirTraf model
- Objective of this study
- Significant aspects of AirTraf 2.0
- Contents of the paper.

The concrete modifications are as follows:
− We deleted some redundant sentences from the introduction (please see the replies to the referee comments (4), (8) and (15)).
− We deleted one paragraph from the introduction (please see the reply to the referee comment (18)).
− We rewrote the text: on page 4 line 3, "Here, we  **mention the importance of** the variety of the routing options."

In addition, we modified the text to make clear what's new in this work and why AirTraf 2.0 is important, and to emphasize its advantage, compared to other models as follows:
− We added the text to the introduction (please see the reply to the referee comment (28)).
− We rewrote the text: on page 3 lines 27-29, "This paper presents a technical description of **the new version of the submodel** AirTraf **2.0** . The simple aircraft routing options of great circle (minimum flight distance) and flight time (minimum time) were developed in **the previous version of** AirTraf 1.0 (Yamashita et al., 2016).  **In AirTraf 2.0**, seven new aircraft routing options have been introduced…."
− We added the word "2.0" to emphasize the new development in AirTraf 2.0: on page 4 line 3, "Various

routing options have been made available in AirTraf **2.0**, because.…"

**c) Methods: clarifying the improvements of AirTraf 2.0 over AirTraf 1.0:**
As the referee noted, AirTraf 2.0 builds on the previous version of AirTraf 1.0. AirTraf is a comprehensive model to enable air traffic simulations on-line in the chemistry-climate model EMAC. In AirTraf 1.0 (Yamashita et al., 2016), we developed the basic modules, including the main structure of trajectory calculations and the optimizer module for flight trajectory optimization. We had also introduced two simple routing options (the great circle and the time-optimal options) to verify, whether the whole system and the optimization module work correctly.

For our ultimate aims described on page 3 line 19, we have expanded the model framework substantially to include seven new routing options with respect to different optimization objectives. We highlight the key changes (i.e. what is new in AirTraf 2.0) in Fig. 1 (on page 22 in the caption, "updates from AirTraf 1.0 are highlighted by red texts and arrows") and in Tables 1 and 2 (on pages 25-26 in the caption, "The column 'New in V2.0' denotes parameters/properties newly introduced in AirTraf 2.0"). We believe that they are useful for readers to recognize the new changes. To show the new changes more clearly, we added the text as follows:

− On page 5 line 2, "The present version is based on the model components of AirTraf 1.0tus, this section outlines them **(updates from AirTraf 1.0 are highlighted in Fig. 1)**."

− On page 5 line 7, "Table 1 lists the relevant data of an A330-301 aircraft and constant parameters used in AirTraf 2.0 **(the new parameters are listed in Table 1)**."

− On page 5 line 28, "The first step finds an optimum flight trajectory for a selected routing option by using the aircraft routing module (Fig. 1, light green)**, in which the seven new routing options are introduced in AirTraf 2.0**."

− On page 6 line 2, "In AirTraf 2.0, 15 new properties are calculated**, as highlighted in Table 2**."

− On page 6 line 5, "… at the departure time of the flight. **The methodologies of the fuel-emissions calculation module developed in AirTraf 1.0 are expanded in AirTraf 2.0.** Details of the fuel-emissions- calculation module  and its reliability have been reported.…"

− On page 6 line 13, "… are gathered along the flight segments (Table 2)**; the global fields of PCC$_{dist}$ and ATR20s are newly calculated by AirTraf 2.0**."

− On page 7 line 15, "**In AirTraf 2.0, S**even new objective functions were developed.…"

**d) Extension of the results section:**
We analyzed simulation results in more detail and additionally rewrote the text for Sect. 3.2 and 3.3. Please see the reply to the referee comment (24).

**e) Language improvements:**
To improve the writing and language, we rechecked and modified the text, redundant words and sentences, articles, and consistency of wording in the manuscript. The referee comments (1), (2) and (4) also pointed out this issue, and thus we replied to them in the corresponding sections. We list here other modifications:

− On page 3 line 6, " **The** benefits of  **the climate-optimized routing** were investigated.…"

− On page 3 line 8, "… pulse AGTP values for three  time horizons.…"

− On page 3 line 10, "… for the medium-term climate goal of 50 years.…"

− On page 3 line 34, "… are referred to simply as, e.g. the "fuel option")."

− On page 10 line 14, "… whereas the trajectories **of the westbound flights** are shifted northward ."

− On page 10 line 26, "… decrease the respective objects (target measures)  **which should** be minimized.…"

− On page 10 line 29, "… Table 4 lists a summary of  **nine** performance measures of.…"

− On page 11 line 26, "… which offers additional aircraft routing options for defining overall target functions for  **the flight** trajectory optimization."

− On page 11 line 29, "The quantitative values of the changes in **the** performance measures vary.…"

− On page 24 in the caption of Fig. 3, "… climate impact indicated by ATR20total  **during the**

**day** (from December 1, 2015 00:00:00 to December 2, 2015 00:00:00 UTC)."

**f) Modification of short sentences:**
We rewrote the three short sentences as follows:
− On page 4 line 5, "The time option is useful for delay recovery. **Because** elays cause costs to airlines  pilots are often forced to temporarily use the time option during a flight to maintain flight schedules.…"
− On page 4 line 11, "AirTraf enables analyzing those subjects." We modified the sentence in reply to the referee comment (28).
− On page 5 line 3, "Thus, this section outlines them." We modified the sentence in reply to the "Methods: clarifying the improvements of AirTraf 2.0 over AirTraf 1.0" described above.

**g) Explanation of terminologies:**
Some terminologies related to the genetic algorithm optimization are used in the present manuscript. We added the explanations to the words and rewrote the text:
− On page 6 line 29, "… and creates an initial **"population,"** which  **represents** a random set of solutions**.** **T**he population size is set by $n_p$ **and ARMOGA starts its search with the solutions**. An evaluation function $f$ **(called an objective function)** is defined, depending on a selected routing option.…"
− On page 7 line 4, "… the stochastic universal sampling selection (Baker, 1985) was used for the selection operator **to pick two solutions (parent solutions) from the population**; the Blend crossover operator (BLX-alpha; Eshelman, 1993) was applied to the  **parent solutions** to create new solutions (child solutions) ; the revised polynomial mutation operator (Deb and Agrawal, 1999) was used to add a disturbance to the child solutions. When  **those** process**es**  **are** iterated for a number of generations (**the term "generation" represents one iteration of ARMOGA;** this is set by $n_g$).…"
− We changed the word "RI" into "RF" (please see the reply to the referee comment (6)).

**h) Formula improvements:**
We added the definitions (equations) of the five ATR20s to the revised manuscript. Please see the reply to the referee comment (21). In addition, we added explanations on Eq. (A5), which is the algorithmic Climate Change Functions of contrails (aCCF$_{contrail}$) to the Appendix. Please see the reply to the referee comment (22).

**i) Modification of references:**
We modified the wrong references as follows:
− On page 3 line 31, "…Yin et al., 2018**a**.…"
− On page 8 line 9, "Yin et al. (2018a) .…"
− On page 9 line 17, "…Yin et al., 2018**a**.…"
− On page 15 line 13, "… published by Van Manen (2017)**, Yin et al. (2018b),** and Van Manen and Grewe (2019); the aCCF of contrails is described by  Yin et al. (manuscript in preparation, 2019)."

- **Selected specific comments:**
  (1) Title: suggest removing "Various". Makes it seem vague.

  Reply: Thank you very much. We removed the word "Various" from the title, and rewrote the title as "**Newly developed** aircraft routing options for air traffic simulation in the chemistry-climate model EMAC 2.53: AirTraf 2.0".

- (2) Abstract: Line 1: Add "the" before "climate impact of aviation.…"

  Reply: We added the word "the" to the revised manuscript: on page 1 line 1 in Abstract, "… **the** **c**limate impact of aviation.…"

- (3) Abstract: Line 6-9: unclear, I don't really understand what the important result here is.

  Reply: We rewrote the text to highlight the messages: on page 1 lines 6-9, "The results clearly show that **AirTraf 2.0 can find** the **different** famil**ies** of optimum flight trajectories (three-dimensional)  **for specific routing options; those trajectories minimize the corresponding objective functions successfully.**  The minimum cost option  **lies** between the minimum time and the minimum fuel  **options. Thus, aircraft operating costs are minimized by taking the best compromise between flight time and fuel use.**"

- (4) Pg1: Line 16: The sentence starting with "the aviation sector is not" is redundant as you've just said that aviation contributes only 5% total climate impact.

  Reply: Thank you very much. We removed the sentence: on page 1 line 16, " However, the aviation's contribution.…" Related to this, we added the word "only" to the revised manuscript: on page 1 line 15, "Nowadays the global aviation contributes **only** about 5 % to the anthropogenic climate impact.…"

- (5) Pg1: Line 23: a more up-to-date reference would be the Brasseur et al. 2016 paper in BAMS.

  Reply: Thank you very much. We referred to the paper in the revised manuscript: on page 1 line 23, "…(Wuebbles et al., 2007; Lee et al., 2009; **Brasseur et al., 2016**)". This paper is listed in the present References.

- (6) Pg2: Line 1: I don't understand the rationale behind introducing the terminology radiative impact (RI) instead of keeping well-established radiative forcing (RF). This is confusing and adds nothing to the paper. Please explain or change.

  Reply: Thank you very much. We believe that the term "radiative impact" is the more general term, and the two sentences, starting from page 1 line 23 "These effects change ... a radiative impact (RI). The RI potentially ... through temperature changes" describe the general mechanism of climate change. Thus, the term "radiative impact" would be appropriate. On the other hand, Lee et al. (2009) and other literature use the term "RF" to report those figures (in $mW/m^{-2}$). By following the referee comment, the best modification would be to remove the abbreviation "RI" in the two sentences, and to use the word "radiative forcing (RF)" in other sentences. Finally, we rewrote the text: from page 1 line 23 to page 2 line 4, "… cause  radiative impact . The  **radiative impact** potentially drives the climate system into a new state of equilibrium through temperature changes. Lee et al. (2009) stated that the $CO_2$ emission has the main impact and that the estimated  **radiative forcing (RF)** of aviation $CO_2$ in 2005 was 28.0 $mWm^{-2}$ (15.2−40.8 $mWm^{-2}$, 90 % likelihood range). The non-$CO_2$ emissions and the induced clouds also have a large effect on  **RFs**; for example, the estimated  **RFs** in 2005.…"

- (7) Pg2: Line 5: there are number of more recent studies showing higher contrail-cirrus forcing, reflecting more recent emission inventories. One example is the 2016 paper by Bock and Burkhardt in JGR-A. Such work should be reflected.

  Reply: Thank you very much. We referred to the three recent papers here and rewrote the sentence: on page 2 line 4, "… the estimated  **RFs** in 2005 for total $NO_x$ and for persistent linear contrails were 12.6 $mWm^{-2}$ (3.8−15.7 $mWm^{-2}$, 90 % likelihood range) and 11.8 $mWm^{-2}$ (5.4−25.6 $mWm^{-2}$, 90 % likelihood range), respectively (Lee et al., 2009). **In particular, the radiative impact of contrails remains uncertain and recent studies report higher RF. Burkhardt and Kärcher (2011) estimated the contrail cirrus RF of 37.5 $mWm^{-2}$ for the year 2002; Schumann et al. (2015) reported the RF of 63 $mWm^{-2}$ for the year 2006; and**

**Bock and Burkhardt estimated the RF of 56 mWm$^{-2}$ for the year 2006.**"

Related to this, we added the three papers to the References:
− On page 17 line 8, "**Bock, L., and Burkhardt, L.: Reassessing properties and radiative forcing of contrail cirrus using a climate model, Journal of Geophysical Research: Atmospheres, 121, 16, 9717−9736, https://doi.org/10.1002/2016JD025112, 2016.**"
− On page 17 line 13, "**Burkhardt, U., and Kärcher, B.: Global radiative forcing from contrail cirrus, Nature Clim Change, 1, 54–58, https://doi.org/10.1038/nclimate1068, 2011.**"
− On page 20 line 34, "**Schumann, U., Penner, J. E., Chen, Y., Zhou, C., and Graf, K.: Dehydration effects from contrails in a coupled contrail–climate model, Atmos. Chem. Phys., 15, 11179–11199, https://doi.org/10.5194/acp-15-11179-2015, 2015.**"

- (8) Pg2: Line 6: "Here the difference between time scales (…)": suggest removing, no point in telling the reader what you will tell them next.

  Reply: We removed the sentence and rewrote the text "As for time scales of their impacts" in context: on page 2 line 6, " **As for time scales of their impacts**.…"

- (9) Pg2 Line 6: "The emitted $CO_2$ (…)" – this is not precise; the emitted $CO_2$ does not have century-long timescale, the perturbation does.

  Reply: We rewrote the sentence: on page 2 line 6, " **the emitted $CO_2$ becomes uniformly mixed in the whole atmosphere and its perturbation remains for millennia.**" This modification is related to our reply to the minor comment (2) of the referee #2.

- (10) Pg2: Line 7: "the impact is proportional to (…)": this may be true for emission, and perhaps even for RF, but when approximating fuel with temperature impact or other climate change seems doubtful.

  Reply: As the referee pointed out, the sentence is not precise. Actually, this sentence does not give any necessary information. Thus, we removed the sentence: on page 2 line 7, "… ."

- (11) Pg2: Line 10: the recent work by Lund et al. 2017 ESD include all components and show how this translates into temperature impacts. Could be a useful references.

  Reply: Thank you very much. We referred to the paper in the revised manuscript: on page 2 line 10, "… Mannstein et al., 2005; Gauss et al., 2006; Grewe and Stenke, 2008; Frömming et al., 2012; Brasseur et al., 2016; **Lund et al., 2017**)."

  Related to this, we added this paper to the References: on page 19 line 26, "**Lund, M. T., Aamaas, B., Berntsen, T., Bock, L., Burkhardt, U., Fuglestvedt, J. S., and Shine, K. P.: Emission metrics for quantifying regional climate impacts of aviation, Earth Syst. Dynam., 8, 547–563, https://doi.org/10.5194/esd-8-547-2017, 2017.**"

- (12) Pg2: Line 17: Why is climate-optimized routing limited to the present-day fleet?

  Reply: The climate-optimized routing is **not** limited to the present-day fleet. What we want to say here is that the climate-optimized routing is **immediately** applicable to the fleet, which airlines currently operate. Although technological measures (e.g. efficient engines, new aircraft) can significantly reduce the aviation climate impact, it takes a long time for airlines to introduce such new technological measures. To make clear the meaning of the sentence, we rewrote the text: on page 2 line 17, "The climate-optimized routing is

**immediately** applicable to present airline fleets, whereas other, more technical measures require several years before implementation."

- (13) Pg2: Line 22: because of the long residence time of $CO_2$, its impact is the same regardless of location of emission. Please be more precise.

  Reply: Thank you very much. We rewrote the sentence more precisely: on page 2 line 21, "… Frömming et al. (2013) and Grewe et al. (2014b) developed Climate Cost Functions (CCFs) for the climate-optimized routing.  **They calculated global-average RFs resulting from local unit emissions ($CO_2$, $NO_x$, $H_2O$ and contrails) over the north-Atlantic for typical weather patterns by using the ECHAM5/MESSy Atmospheric Chemistry model EMAC (Jöckel et al 2010, 2016). Those RFs were used to calculate the global and temporal average near-surface temperature response over 20 years, which describe the climate impacts (i.e. future temperature changes) caused by those emissions on a per unit basis. The resulting data set is called the Climate Cost Functions (CCFs). The CCFs describe the climate impact which is induced by aviation's $CO_2$ and non-$CO_2$ effects ($H_2O$, ozone, methane, ozone originating from methane changes, and contrails including the spread into contrail-cirrus); and the CCFs of those effects except $CO_2$ are a function of geographic location, altitude and time. Because of the long residence time of $CO_2$, its impact is the same regardless of location, altitude and time of emission. The obtained CCFs can be used as a measure of the climate impact of aviation and form the basis for the climate-optimized routing.**"

  Related to this, we rewrote the following text, because the modified sentences described above refer to the word "EMAC" for the first time in the revised manuscript: on page 3 line 22, "… developed as one of the submodels of  **EMAC**.…"

- (14) Pg2: Line 22: Please add a more detailed definition of CCS as the reader needs this later on.

  Reply: Thank you very much. We added the details of the Climate Cost Functions (CCFs) to the revised manuscript. Please see the reply to the comment (13).

- (15) Pg2: Line 24: Another strange sentence to suddenly introduce here instead of adding above when listing aviation non-$CO_2$ effects.

  Reply: We deleted this sentence: on page 2 line 24, "". In addition, we added this description to the list of non-$CO_2$ effects: on page 2 line 23, "… and non-$CO_2$ effects ($H_2O$, ozone, methane, **ozone originating from methane changes**, and contrails.…" Please see the reply to the comment (13).

- (16) Pg2: Line 29: what about trade-offs between e.g., contrail avoidance and increased fuel use?

  Reply: As the referee pointed out, the trade-off between contrail avoidance and increased fuel use is also an important subject. Actually, this is the reason why we develop many routing options in AirTraf 2.0; one can analyze the trade-off by using AirTraf 2.0. This point is described in the paragraph (on page 4 lines 3-13). On the other hand, the paragraph (on page 2 lines 20-33) focuses on the "climate-optimized routing," and thus we did not mention trade-offs between other routing strategies there.

  To emphasize the importance of analyzing other trade-offs, we rewrote the text: on page 4 line 9, "Moreover, conflicting scenarios (trade-offs) between different routing strategies have been studied; for example, **avoiding contrail formation generally increases fuel use and $CO_2$ emissions.** Irvine et al. (2014) assessed  **the** trade-off between  **contrail avoidance** and  **increased** $CO_2$ emission (~  **increased** fuel use) for a single flight."

- (17) Pg3: Line 2: Presumably this is global-mean temperature response? Please specify.

  Reply: The referee is right. This represents the global-mean temperature response. Unfortunately, this part is deleted by following the referee comment (18). Please see the reply to the comment (18) below.

- (18) Pg3: Line 5: what about the other way around, does a cost-optimized route increase climate impact?

  Reply: In this paragraph (on page 2 line 34 - on page 3 line 5), Lührs et al. (2016) clearly show a trade-off between climate impact and economic cost. Thus, as the referee pointed out, one can say that the cost-optimized route increases climate impact with a decrease in cost, compared to the climate-optimized route.

  On the other hand, we deleted this paragraph to shorten the introduction by following the referee's general comment. This paragraph introduces the study of Lührs et al. (2016) which clearly shows the trade-off between climate impact and economic costs; however, the previous paragraph (on page 2 lines 20-33) has already introduced two studies to show the same trade-off on the basis of the same climate metrics CCFs. Thus, this paragraph would be redundant. On page 2 line 34, "~~Lührs et al. (2016) performed a flight trajectory optimization for nine sample trans-Atlantic routes for a specific weather pattern in winter by the Trajectory Optimization Module (TOM). The trajectories were optimized for economic cost (expressed by the cash operating cost (COC; Liebeck et al., 1995; see Sect. 2.5.6), which is commonly used as a criterion for airline economics) and for climate impact (measured as average temperature response estimated by integrating the CCFs). The results showed that the climate-optimal route differed from the cost-optimal route. The climate-optimum trajectory (3D-optimized trajectory in lateral and vertical) decreased the climate impact by about 45 % over that of the economical route, whereas it increased COC by 2 %. Thus, the climate impact drastically decreased with a small increase of economic cost.~~"

  Related to this, we moved the reference "Lührs et al. (2016)" from the current position (on page 2 line 34) to another position: on page 2 line 21, "… 2013; Søvde et al., 2014**; Lührs et al., 2016**).…" In addition, we rewrote the text, because the deleted paragraph refers to the word "COC" for the first time in the present manuscript: on page 3 line 29, "… simple operating cost (SOC), **cash operating cost** (COC), and climate impact.…"

- (19) Pg3: Line 6: do you mean using different emission metrics, of which AGTP is one? And which other metrics do you find in the literature? Here you only describe one approach. (from here on I do not list language issues, but note that there are a number of them also in the next pages...)

  Reply: Yes. We believe that it is important to show that the benefit of the climate-optimized routing is confirmed on the basis of different climate assessment metrics (AGTP is one of them). On page 3 line 6, Ng et al. (2014) clarified the benefit by using the three AGTP values for the short (25 years), medium (50 years) and long-term (100 years) climate goals. As we only described the results for the medium-term climate goal in the present manuscript, we added the text below to the revised manuscript.

  As the referee pointed out, there are other climate metrics. For example, Grewe et al. (2014a) compared the trade-off between economic costs and climate impact from one-day trans-Atlantic air traffic simulations with respect to three climate metrics. The results indicated that all metrics show a similar trade-off between economic costs and climate impact. We believe that this information would be useful for readers, and thus we added this information to the revised manuscript.

  Finally, we rewrote the text: on page 3 line 14, "… between climate impact and economic cost**; this trade-off was also found for the short-term (25 years) and long-term (100 years) climate goals. Grewe et al. (2014a) compared the trade-off between economic costs and climate impact from the one-day trans-Atlantic air traffic simulations described above with respect to three climate metrics: the average temperature response with future increasing emissions (F-ATR20) and the absolute global warming**

**potential with pulse emissions at a 20 year time horizon (P-AGWP20) for short-term climate impacts, and P-AGWP100 (time horizon of 100 years) for long-term climate impacts. The trade-offs obtained with the three metrics were very similar.**"

Concerning language issues, we rechecked the manuscript and added some modifications to the revised manuscript. We list them in the reply to the referee's general comment.

- (20) Pg8: Section 2.5.4: The treatment of contrail-cirrus is quite essential for routing strategies and I would like to see some more details of how this is done and what the limitations are (e.g., natural cloud suppression, life cycle etc.) here, not just a reference to earlier work.

Reply: Thank you very much. We rewrote the paragraph to describe more details of how this routing option is made and its limitations: on page 8 line 9, "Yin et al. (2018a)  developed the routing option  **to avoid contrail formations by using the submodel CONTRAIL (version 1.0; Frömming et al., 2014), which calculates the potential persistent contrail cirrus coverage Potcov (Ponater et al., 2002; Burkhardt et al., 2008; Burkhardt and Kärcher, 2009; Grewe et al. 2014b) within an EMAC grid box.**  **The Potcov represents the fraction of the grid box, which can be maximally covered by contrails under the simulated atmospheric condition. The threshold for contrail formation is determined from a parameterization scheme based on the thermodynamic theory of contrails, i.e., the Schmidt-Appleman theory (Schmidt, 1941; Appleman, 1953; Schumann, 1996). In the CONTRAIL submodel, Potcov indicates the difference between the maximum possible coverage of both, contrails and cirrus, and the coverage of natural cirrus alone; values of Potcov along the waypoints are taken from the nearest grid box (Table 2). With that, we define a contrail distance ($PCC_{dist}$) in km(contrail) as Potcov multiplied by the flight distance in km. The corresponding routing option minimizes the total contrail distance of a flight and thus the** objective function  **is formulated as**:…."

In addition, we rewrote the text: on page 8 lines 14-19 in the same paragraph, "~~PCC$_{dist,i}$ is calculated by using the potential contrail coverage Potcov (Ponater et al., 2002; Burkhardt et al., 2008; Burkhardt and Kärcher, 2009; details of Potcov have been reported by Frömming et al., 2014). The Potcov represents fractional areas in which contrails can maximally occur under a given atmospheric condition. The Potcov is calculated by the submodel CONTRAIL (version 1.0; Frömming et al., 2014), using a parameterization scheme based on the thermodynamic theory of contrails, i.e., the Schmidt-Appleman theory (Schmidt, 1941; Appleman, 1953; Schumann, 1996)~~ **Note that the objective function is formulated in the simple form to consider only the contrail distance. Thus, further physical processes such as contrail spreading, changes in contrail coverage area, contrail lifetime, and the contrail radiative forcing are not included**."

- (21) Pg9: Line 20: ATR20 needs a definition. Is it calculated based on input of RF? What is assumed for contrail-cirrus properties?

Reply: Thank you very much. We replied the three referee comments, respectively.

["ATR20 needs a definition."]
Table 2 (on page 26) included the definitions (equations) of the five ATR20s; however, as the referee noted, those definitions are important information for readers to understand the climate impact routing option. Thus, we moved those equations from Table 2 to Sect. 2.5.7 and rewrote the text as follows:

− On page 26 in Table 2 (the second group divided by rows), " **See Eq. (8)**;  **See Eq. (9)**;  **See Eq. (10)**;  **See Eq. (11)**;  **See Eq. (12)**;  **See Eq. (13)**."

− On page 9 line 20, "… ATR20s of ozone, methane, water vapour, $CO_2$, and contrails are estimated on a per unit

basis **by**

$ATR20_{O3,i} = aCCF_{O3,i} \, NO_{x,i} \times 10^{-3}$ **(8)**
$ATR20_{CH4,i} = aCCF_{CH4,i} \, NO_{x,i} \times 10^{-3}$ **(9)**
$ATR20_{H2O,i} = aCCF_{H2O,i} \, FUEL_i$ **(10)**
$ATR20_{CO2,i} = aCCF_{CO2} \, FUEL_i$ **(11)**
$ATR20_{contrail,i} = aCCF_{contrail,i} \, PCC_{dist,i}$ **(12)**

**where the respective aCCF values of ozone, methane, water vapour, $CO_2$, and contrails are given as flight properties at the $i^{th}$ waypoint.** These five ATR20s are.…"

− On page 9 line 24, "$ATR20_{total,i} = …$, **(13)**."
− On page 9 line 25, "$f = …$, **(14)**."

["Is it calculated based on input of RF?"]
In AirTraf 2.0, ATR20s are calculated for the climate-optimized routing by using the algorithmic Climate Change Functions (aCCFs) of ozone, methane, water vapour, $CO_2$, and contrails (shown in the Appendix), for which RF is not used as an input parameter. However, the aCCFs are approximation functions based on regression analyses for the CCFs data set (this point is described on page 9 line 18). As we reply to the referee comment (13), the CCFs data set was obtained from detailed EMAC model simulations including RF calculations (for contrails, the calculated RF data set was obtained in a different way; details are described in the "What is assumed for contrail-cirrus properties?" below); the CCFs data set describes the climate impact which is induced by ozone (plus ozone originating from methane changes), methane, $H_2O$, $CO_2$, and contrails. Thus, the aCCFs approximately express the climate impact (ATR20) by taking radiative impacts into account.

["What is assumed for contrail-cirrus properties?"]
The ATR20 of contrails is calculated by using the approximation function of $aCCF_{contrail}$ in AirTraf 2.0; the $aCCF_{contrail}$ was created from contrail RF calculations based on the ERA-Interim reanalysis data and contrail trajectory data. To reply to this referee comment, let us explain the derivation of $aCCF_{contrail}$ briefly. First, the contrail RF data set was calculated following these steps:

(a) Lagrangian trajectories (air parcels) were computed by using the ERA-Interim reanalysis data (the methodology is described by Irvine et al., 2014); the trajectories were initialized over the north Atlantic (1 degree horizontal spacing) at three vertical levels (300, 250 and 200 hPa) in winters of 1994, 1995 and 2003. The contrail lifetime was calculated by analyzing each of the trajectories to see how long the conditions persisted for: relative humidity with respect to ice above 98 % and a temperature below 235 K.

(b) Contrail properties were calculated along the trajectories by following Schumann et al. (2017), where an effective radius for contrail cirrus ice particle was set to 23 microns described by Schumann et al. (2011). The contrail optical depth was calculated by a simple formula for the extinction coefficient (Unterstrasser and Gierens, 2010), where the initial contrail depth was set to 200 m (Grewe et al., 2014).

(c) The long-wave and short-wave RFs were calculated from the trajectory data by using the parametric equations described by Schumann et al. (2012). The area covered by each contrail was assumed constant along the trajectory. By taking values from Grewe et al. (2014), we used a contrail width of 200 m, and a contrail length of the square root of the grid box area (1 degree by 1 degree grid). The net RF was calculated for each contrail and was converted to a global-mean value by following Grewe et al. (2014). The contrail RF data set was obtained, in which the lifetime of contrails ranges from 3 to 48 hours.

The $aCCF_{contrail}$ was derived based on regression analyses for the RF data set. The methodology was based on that used by van Manen and Grewe (2019) to derive the other aCCFs for ozone, methane and water vapour. For the regression analyses, a constraint on deriving $aCCF_{contrail}$ was that only meteorological information available at the time of flight can be used. In addition, we restricted the calculation to conventional meteorological data, so that $aCCF_{contrail}$ was simple to implement. This means, for example, no information on

the contrail lifetime could be used, because this is not something which can be estimated a priori from meteorological data. Since a lifetime was required to be input to the net RF calculations, we chose a contrail lifetime of six hours for all contrails, because 92 % of contrails have a lifetime up to six hours in the data set. Night-time and day-time contrails were analyzed separately. The night-time contrails referred to contrails with their entire (six hours) lifetime occurring at night; the day-time contrails referred to contrails which existed only during daylight hours and those which had part of their lifetime during the day. The obtained aCCF$_{contrail}$ (Eq. (A5) on page 15 in the Appendix) was converted from RF to ATR20 by multiplying a factor of 0.114 (provided by Katrin Dahlmann, DLR).

The derived aCCF$_{contrail}$ has been assessed by plotting the original net RF with the RF calculated by using aCCF$_{contrail}$. In addition, the performance of aCCF$_{contrail}$ has been assessed against the rest of the contrails with lifetimes of 3 to 48 hours in the data set. The Spearman's rank correlation coefficient and the ability to correctly predict the sign of the forcing were examined. For day-time contrails with lifetimes of 6 hours, for example, the coefficient was R = 0.86, and the ability (in percentage) was 88 %; for those with lifetimes of 12 hours, R = 0.83 and 78 % were obtained. These results provide the confidence in the use of aCCF$_{contrail}$ in the aircraft routing decision.

Here, we would like to make clear that the literature, which is given on page 9 lines 17-18, describes how to develop aCCFs from the CCFs data set, and their limitations in detail. The aCCFs are calculated online in EMAC by another submodel named ACCF in MESSy (version 2.54), and thus the AirTraf submodel uses the ACCF submodel for the climate routing option. In addition, the detailed description of the CCFs data set was added to the revised manuscript by following the referee comment (13).

Finally, we rewrote the text to show the relation between the CCFs data and the aCCFs more clearly: on page 9 line 18, "The aCCFs are approximation functions based on regression analyses for the  CCFs data **set, which was obtained from detailed EMAC model simulations including radiative impacts** (see Sect. 1); **the CCFs data set for contrails was exceptionally obtained from contrail RF calculations based on the European Centre for Medium-Range Weather Forecasts (ECMWF) Re-Analysis Interim (ERA-Interim) data (Dee et al. 2011) and contrail trajectory data (Yin et al. (manuscript in preparation, 2019); the definition of the aCCFs**  **is provided in the Appendix and examples are shown in Fig. S1 in the Supplementary material).** The aCCFs represent.…"

Related to this, we added this paper to the References:
− On page 17 line 32, "**Dee, D. P., Uppala, S. M., Simmons, A. J., Berrisford, P., Poli, P., Kobayashi, S., Andrae, U., Balmaseda, M. A., Balsamo, G., Bauer, P., Bechtold, P., Beljaars, A. C. M., van de Berg, L., Bidlot, J., Bormann, N., Delsol, C., Dragani, R., Fuentes, M., Geer, A. J., Haimberger, L., Healy, S. B., Hersbach, H., Hólm, E. V., Isaksen, L., Kållberg, P., Köhler, M., Matricardi, M., McNally, A. P., Monge-Sanz, B. M., Morcrette, J-J., Park, B-K., Peubey, C., de Rosnay, P., Tavolato, C., Thepaut, J-N., Vitart, F.: The ERA-Interim reanalysis: configuration and performance of the data assimilation system., Q. J. R. Meteorol. Soc., 137, 553–597, https://doi.org/10.1002/qj.828, 2011.**"

References:
Grewe, V., Frömming, C., Matthes, S., Brinkop, S., Ponater, M., Dietmüller, S., Jöckel, P., Garny, H., Tsati, E., Dahlmann, K., et al.: Aircraft routing with minimal climate impact: the REACT4C climate cost function modelling approach (V1.0), Geoscientific Model Development, 7, 175–201, https://doi.org/10.5194/gmd-7-175-2014, 2014.

Irvine, E. A., Hoskins, B. J., Shine, K. P.: A Lagrangian analysis of ice-supersaturated air over the North Atlantic, J. Geophys. Res., 119, 1, 90–100, https://doi.org/10.1002/2013JD020251, 2014.

Schumann, U., Baumann, R., Baumgardner, D., Bedka, S. T., Duda, D. P., Freudenthaler, V., Gayet, J.-F., Heymsfield, A. J., Minnis, P., Quante, M., Raschke, E., Schlager, H., Vázquez-Navarro, M., Voigt, C., and Wang, Z.: Properties of individual contrails: a compilation of observations and some comparisons, Atmos.

Chem. Phys., 17, 403–438, https://doi.org/10.5194/acp-17-403-2017, 2017.

Schumann, U., Mayer, B., Graf, K., and Mannstein, H.: A Parametric Radiative Forcing Model for Contrail Cirrus, Journal of Applied Meteorology and Climatology, 51, 6, https://doi.org/10.1175/JAMC-D-11-0242.1, 1391–1406, 2012.

Schumann, U., Mayer, B., Gierens, K., Unterstrasser, S., Jessberger, P., Petzold, A., Voigt, C., and Gayet, J-F.: Effective Radius of Ice Particles in Cirrus and Contrails, Journal of the Atmospheric Sciences, 68, 2, 300–321, https://doi.org/10.1175/2010JAS3562.1, 2011.

Unterstrasser, S. and Gierens, K.: Numerical simulations of contrail-to-cirrus transition – Part 1: An extensive parametric study, Atmos. Chem. Phys., 10, 2017–2036, https://doi.org/10.5194/acp-10-2017-2010, 2010.

Van Manen, J. and Grewe, V.: Algorithmic climate change functions for the use in eco-efficient flight planning, Transportation Research Part D: Transport and Environment, 67, 388–405, https://doi.org/10.1016/j.trd.2018.12.016, 2019.

- (22) Pg9: Line 26: But ATR20 is an average over 20 years? How can values be negative when the overall contrail-cirrus effect is a warming? Perhaps related to the above comment…

  Reply: Yes. ATR20 represents an average over 20 years. In AirTraf 2.0, the ATR20 of contrails is calculated by using the approximation function of $aCCF_{contrail}$. The $aCCF_{contrail}$ consists of two formulas for the day-time and night-time contrails, as shown in Eq. (A5) on page 15 in the Appendix. The $aCCF_{contrail}$ for the day-time contrails can take positive and negative values, depending on the outgoing long-wave radiation (OLR) (the threshold is $-193.18$ $Wm^{-2}$), whereas the $aCCF_{contrail}$ for the night-time contrails takes positive values.

  In the revised manuscript, we rewrote the text to make clear the points described above:
  − On page 9 line 26, "… $ATR20_{contrail,i}$ can take positive and negative values, because  **the $aCCF_{contrail}$ consists of two formulas for** the day-time and night-time contrail effects **(see Eqs. (12) and (A5) in the Appendix).**"
  − On page 16 line 6 in the Appendix, "… calculated by AirClim. **The $aCCF_{contrail}$ for the night-time contrails takes positive values; if the temperature is less than 201 K, $aCCF_{contrail}$ for the night-time contrails is set to zero. The $aCCF_{contrail}$ for the day-time contrails can take positive and negative values, depending on the OLR (the threshold is $-193.18$ $Wm^{-2}$).**" The rewriting highlighted by blue texts comes from our reply to the major comment (2) (starting with "Firstly, Eq. (A5) assumes") of the referee #2.

  This referee comment is related to the referee comment (21). We describe how $ATR20_{contrail}$ is calculated in the AirTraf submodel, and how $aCCF_{contrail}$ was created in the reply to the referee comment (21).

- (23) Pg10: Line 3: how sensitive are results and conclusions to the running of only one day? E.g., dependence on meteorological conditions that day?

  Reply: We acknowledge that the simulation results depend on the atmospheric conditions of the target day. If we perform an AirTraf simulation with the same flight plan for another day, we obtain different optimized trajectories and performance measures. Thus, we clarified this point: on page 11 line 22, "Note that this performance is a narrow result obtained using AirTraf 2.0 under the specific conditions (e.g., the simulations were carried out with the 103 north-Atlantic flights on December 1, 2015…"; and on page 11 line 29, "The quantitative values of the changes in performance measures vary, depending on different methodologies, atmospheric conditions.…"

  We believe that it is an important point to examine whether the findings described in the Conclusions (e.g. the trade-off between the cost and the climate impact) are common under any atmospheric conditions. Actually, this is our next study. Recently, Yamashita et al. (2020) examined this for representative weather types over

the North Atlantic by using EMAC with AirTraf 2.0.

To emphasize the importance of the point, we added the text: on page 15 line 5, "The integration of AirTraf into EMAC allows one to optimize **flight trajectories** and to study  **aircraft routings** under historical, present-day and future conditions of the climate system. **We acknowledge that the simulation results depend on the atmospheric conditions of the target day. Thus, it is important to examine whether the findings, e.g., the trade-off between the cost and the climate impact, are common under any atmospheric conditions. Recently, Yamashita et al. (2020) examined this for representative weather types over the North Atlantic by using EMAC with AirTraf 2.0.** Furthermore, the integrated aircraft routing options could be extended to conflicting scenarios.  Yin et al. (2018a)…."

Related to this, we added the literature "Yamashita et al. (2020)" to the References: on page 21 line 25, "**Yamashita, H., Yin, F., Grewe, V., Jöckel, P., Matthes, S., Kern, B., Dahlmann, K., and Frömming, C.: Comparison of various aircraft routing strategies using the air traffic simulation model AirTraf 2.0, 3rd ECATS Conference, Gothenburg, Sweden, 1–4, 2020.**"

- (24) Pg10: Line 11: showing direct results is not a verification of simulations output.

  Reply: Thank you very much. As the referee pointed out, the sentence starting with "To verify" is inappropriate. Section 3 focuses on a demonstration of AirTraf 2.0, and we intend to show the simulation output as an example in Sect. 3.2. On the other hand, Sect. 4 verifies the consistency of the simulation results with literature data. For the appropriate wording, we changed the word "verify" into the "demonstrate": on page 10 line 11, "To  **demonstrate** the simulation output…"; on page 14 line 24, "To  **demonstrate** the submodel AirTraf 2.0, example simulations were carried out.…"

  This referee comment is related to the referee's general comment: "While the discussion section is quite good, the result is only one page out of a 14-page paper, which is not quite convincing." To provide convincing explanations for the simulation output, we analyzed the simulation results in more detail, added the two new figures "**Figure 3**" and "**Figure 5**", and additionally wrote the text as follows:

  [Section 3.2]
  − On page 10 line 18, "… flight altitudes (∼FL410, 12.5 km)**. Figure 3 shows the mean fuel consumption (in kg(fuel)/min$^{-1}$) vs. mean flight altitude (in km) for individual flights for the three routing options.** Because fuel consumption decreases  **as a result of** aerodynamic drag reduction **at high altitudes** (Fichter et al., 2005; Schumann et al., 2011; Yamashita et al., 2016)**, the COC optimum trajectories select the high flight altitudes, as shown in Fig. 3. We acknowledge that limitations of BADA 3 affect the selection of the flight altitudes (the same applies to the fuel, the NO$_x$, the H$_2$O and the SOC options; see Fig. S3 in the Supplementary material). According to Nuic et al. (2010), BADA 3 has a tendency to underestimate aircraft fuel consumption at high altitudes and Mach numbers, as the compressibility effect and wave drag are not modeled. These effects will cause differences in the selection of the flight altitudes.**" This rewriting comes from our reply to the major comment (3) of the referee #2.

  As we add the new figure, we changed the original figure number: on page 10 line 21, "Figure 4 shows …"; on page 10 line 25, "… it is apparent from Fig. 4 …"; and on page 24 in the caption, "Figure 4."

[Figure]

**Figure 3. Mean fuel consumption vs. mean flight altitude for 103 individual flights obtained by the contrail formation, the COC and the climate impact routing options.**

− On page 10 line 22, "We see **from Figs. 4b, 4e and 4h** that the contrail option certainly decreases the contrail  **formation, which is mostly located over northwest Europe and over the east coast of the U.S.;**  **Figs. 4a, 4d and 4g shows that** the COC option  **produces** a narrower fuel distribution than that of the contrail and climate options.  **In addition, Figs. 4c, 4f and 4i show that** the climate option  **decreases the positive values of ATR20$_{total}$ (warming effects) over northwest Europe and over the east coast of the U.S., and produces regionally negative values (cooling effects) near Iceland and over eastern Canada, which result in** the net climate impact reduction ."

[Section 3.3]
− On page 11 line 4, "**The individual routing options are now discussed in turn.** We see from Table 4 that the great circle option has the minimum flight distance **of 660.3 × 10$^3$ km**, whereas this option increases **the** other measures. The time option shows the minimum flight time **of 739.4 h with a large** penalty on fuel use, NO$_x$ emission.…"

− On page 11 line 6, "… (further discussion in Sect. 4). **The fuel option shows the minimum fuel use of 3758.5 ton. Of the nine routing options,** the fuel (and also the H$_2$O), the NO$_x$, the SOC, and the COC options obtain similar values on all the measures (see also Supplement Fig. S4)**:**  these options show decreased fuel use, NO$_x$ and H$_2$O emissions.…"

− On page 11 line 9, "… is considered significant for airline operations and thus is discussed **in more detail** in Sect. 4. The contrail option shows the minimum contrail distance **of 26.3 × 10$^3$ km** and  **the second-lowest** ATR20$_{total}$ **of 3.45 × 10$^{-7}$ K**, whereas the other measures  increase **considerably**. This option allows aircraft to widely detour the potential contrail regions (because no constraint function is used in Eqs. (1) and (5)**; see below for more discussion**). Thus, **the flight distance,** the flight time and the fuel use  increase **drastically**, which results in the increase of NO$_x$ and H$_2$O emissions, SOC, and COC. **In particular,** the contrail option shows the highest  COC **of 5.99 Mil.USD** of the nine routing options."

− On page 11 line 15, "The two options show similar values for all the measures and have the same minimum SOC **of 3.96 Mil.USD** and COC **of 5.35 Mil.USD**.  In fact, the obtained optimum trajectories for  **those**

options are approximately the same (see Figs. 2c, 2d and Supplement Figs. S3k and S3l). **This is because the objective function of the two options is a function of flight time and fuel, as defined in Eqs. (6) and (7). An interesting aspect of their performance measures is that both options do not correspond to the minimum flight time and fuel use (see further discussion in Sect. 4).**"

− On page 11 line 18, "The climate option achieves the minimum $ATR20_{total}$ of $1.96 \times 10^{-7}$ K and  the **second-shortest** contrail distance **of $92.6 \times 10^3$ km**, whereas  **the** other measures **increase**, particularly  **this option shows the second-highest COC of 5.87 Mil.USD**. The present results indicate that the contrail and the climate options considerably reduce the climate impact indicated by $ATR20_{total}$**; however, these options increase COC.**"

− On page 11 line 24, "**Figure 5 shows the contrail distance (in $\times 10^3$ km) vs. $ATR20_{contrail}$ (in $\times 10^{-7}$ K) for individual flights for the contrail, the COC, and the climate options. We see that the contrail option decreases the contrail distance drastically and shows the positive values of $ATR20_{contrail}$ for almost all the flights. On the other hand, the climate option has the longer contrail distances than those of the contrail option (although the climate option achieves the second-shortest total contrail distance, as shown in Table 4) and shows the negative values of $ATR20_{contrail}$ for many flights. These results imply that the contrail option minimizes the overall contrail distance at all times, whereas the climate option actively forms cooling contrails during the day and avoids the formation of warming contrails during the day and night.**" This rewriting comes from our reply to the major comment (4) of the referee #2.

[Figure]

**Figure 5. Contrail distance vs. $ATR20_{contrail}$ for 103 individual flights obtained by the contrail formation, the COC and the climate impact routing options.**

- (25) Pg10: Line 21: over what time frame is the km coverage estimated? Integrated over the 1-day simulations?

Reply: Yes. We integrated the contrail distance [km] of the total 103 flights over the target day. To clarify this point, we rewrote the text: on page 10 line 21, "**T**he global fields of fuel use, contrail distance, and climate impact indicated by $ATR20_{total}$ for the three options **are shown in Fig. 3, where distributions represent sum of all the flights during the day.**"

- (26) Pg11: Line 2-3: this is a very strange argument for correctness.

Reply: We rewrote the text: on page 11 line 2, "These results confirm  **that the new routing options work correctly in AirTraf 2.0**, since we solve a single-objective

minimization problem defined by Eq. (1).…"

- (27) Pg14: Line 10-11: how well does the treatment of contrails work for longer time integrations (in particular decades as mentioned earlier)? Is the potcov based on present day conditions?

  Reply: This is an important point, and this referee comment is related to the referee comments (20), (21) and (22). The climate routing option uses aCCF$_{contrail}$. The aCCF$_{contrail}$ estimates the anticipated climate impact of contrails ATR20$_{contrail}$, which is caused by local contrail formation during the present day, on the basis of the present day conditions including potcov; the calculated impact of contrails is integrated over time. As we reply to the referee comments (20), (21) and (22), aCCF$_{contail}$ (as shown in Eq. (A5) on page 15 in the Appendix) represents the climate impact of contrails, taking into account physical processes of contrails over a longer time period (e.g. contrail lifetime, contrail radiative forcing, etc.). This is because aCCF$_{contrail}$ has been developed from the CCFs data sets obtained from contrail RF calculations based on the ERA-Interim reanalysis data and contrail trajectory data over a longer time period, in which such physical processes of contrails were included.

  We believe that the replies to the referee comments (20), (21) and (22) describe this point in detail; those descriptions were added to the revised manuscript (please see the replies to the referee comments (20), (21) and (22)).

- (28) Pg14: Line 5-10: this type of information would be useful in the introduction.

  Reply: Thank you very much. We moved the information from the current position (on page 14 lines 5-10) to the introduction, and then we rewrote the text as follows:

  − On page 14 lines 4-11, "As discussed above, the many previous studies  **corroborate** the consistency of the AirTraf simulations. ~~Before concluding the discussion, two superior aspects of the AirTraf submodel are emphasized, compared to the simulation models used in the previous studies. First, AirTraf enables an intercomparison for various aircraft routing options all at once, because all the options are integrated. Normally, one or two specific routing options are available for a flight trajectory optimization in other models. Second, AirTraf performs air traffic simulations not under ISA conditions, not under a fixed atmospheric condition for a specific day, but under comprehensive atmospheric conditions which are calculated by the chemistry-climate model EMAC. AirTraf can simulate air traffic for long-term period in EMAC, which enables one to examine effects of aircraft routing strategies on climate impact on a long time scale.~~"

  − On page 4 line 11, "… for a single flight. AirTraf **2.0** enables analyzing those subjects **all at once, because all the options are integrated. Normally, one or two specific routing options are available for a flight trajectory optimization in other models. Another aspect to be emphasized compared to other models is that AirTraf performs air traffic simulations not under International Standard Atmospheric (ISA) conditions, not under a fixed atmospheric condition for a specific day, but under comprehensive atmospheric conditions which are calculated by EMAC; that is, AirTraf can simulate air traffic for long-term periods in EMAC, which enables one to examine effects of aircraft routing strategies on climate impact on a long time scale.** Last but not least, the aCCFs are new proxies.…"

  Related to this, we rewrote the following text, because the modified sentences described above refer to the word "ISA" for the first time in the revised manuscript: on page 12 line 8, "… respectively, under  conditions. A typical single-aisle aircraft.…"

---

## Author Comment (AC2) · 10 Jun 2020

We are grateful to the referee #2 for the very helpful and encouraging comments on the original version of our manuscript. We took all comments into account and rewrote the manuscript accordingly. Here are our replies:

- This paper proposes an updated sub-model in the ECHAM Atmospheric Chemistry model for flight trajectory optimisation and is within the scope of the journal (GMD). The algorithm now enables the flight trajectory to be optimised based on various scenarios, which can assist relevant stakeholders and policymakers to evaluate the tradeoff between economic costs and the overall climate impact. Such a tool is expected to become increasingly important as the focus shifts to minimising aviation's overall environmental impact, including both $CO_2$ and non-$CO_2$ emissions.

    While the work is well structured and written, there are several major aspects in the model that were not adequately addressed and must be significantly improved. Therefore, I believe that major revisions are necessary before this paper is accepted for publication.

    Reply: We thank the referee #2 for these positive comments. We have addressed all the major and minor comments as follows.

- **Major Comments:**
    **1)** [Page 6, Line 12] What is the rationale for selecting a 20-year time horizon for the average temperature response (ATR20)? Given that a proportion of $CO_2$ can remain in the atmosphere for over a millennium [Ref.1], the ATR20 can lead to a large underestimation in the $CO_2$ climate impacts. To overcome this, it is suggested that the authors perform a sensitivity analysis on the reported results by considering the use of different ATR time horizons (i.e. 100 years and 1000 years).

    Reply: As the referee pointed out, a choice of the climate metric is an important issue. Grewe and Dahlmann (2015) pointed out that the different climate metrics, although targeting somehow climate change, provide "different physical quantities measuring climate change and hence they provide answers to different questions." From this viewpoint, we selected the climate metric of ATR20 on the basis of the five steps proposed by Grewe and Dahlmann (2015). First, we posed the detailed question: "what potential reduction in climate impact could be achieved by steadily applying a climate optimizing aircraft routing strategy in the next few decades?" From this objective, we considered a business-as-usual future air traffic scenario for one-day trans-Atlantic flights as a reference, and compared that to a scenario where we daily flew trans-Atlantic routings with a low climate impact. To answer the question, finally we selected an appropriate climate metric: the global and temporal average near-surface temperature response over 20 years after introducing the climate-optimized routing strategy. This metric enables the different climate relevant emissions to be placed on a common scale and thus be directly compared.

    We have performed a sensitivity analysis on the climate metrics in previous research (Grewe et al., 2014). We optimized one-day trans-Atlantic air traffic with respect to three climate metrics: the average temperature response with future increasing emissions (F-ATR20) and the absolute global warming potential with pulse emissions at a 20 year time horizon (P-AGWP20) for short-term climate impacts, and P-AGWP100 (time horizon of 100 years) for long-term climate impacts. The results indicated that the Pareto fronts (optimal relation between climate change and costs) are similar for the three metrics (shown in Fig. 3 of Grewe et al., 2014). For each metric, the relative importance of individual species ($CO_2$, contrails, ozone, methane, and total $NO_x$ ($O_3$ + $CH_4$ + PMO)) for a reduction of the climate impact was also investigated, and all metrics showed a similar pattern (shown in Fig. 4 of Grewe et al., 2014). These results were very robust in terms of dependence from the chosen metric and in terms of the role of individual components. We noticed that if we had adopted the more frequently used pulse-based metrics (e.g. Fuglestvedt et al., 2010), we would have found a much stronger sensitivity of the short-lived effects, e.g., contrails; the more contrast between the short-lived effects, such as contrails, and the long-lived emissions, such as $CO_2$, would also be expected. However, these would not have been the best suited to quantifying the sustained impact of a permanent change in routing strategy on near-term climate change. That is, such metrics are not suitable to answer the aforementioned question.

On the basis of the rationale for selecting ATR20 described above, we applied the metric to the calculated RF, which was obtained from the detailed EMAC model simulations (for contrails, the RF data set was obtained in a different way; details are described in the reply to the referee major comment (2) starting with "Secondly, some contrails" below), and then obtained a relation between locally and temporarily specified emissions and the global-average impact on climate in terms of future temperature changes (ATR20). We call these 4-D response patterns as "climate-change functions (CCFs)." Algorithmic Climate Change Functions (aCCFs) are approximation functions based on regression analyses for the CCFs data set. Thus, aCCFs approximately express the climate impact indicated by ATR20. The aCCFs have already been published as the ACCF submodel in MESSy (version 2.54), and thus the AirTraf submodel uses the ACCF submodel for the climate-optimized routing. We would like to note that the literature, which is given on page 9 lines 17-18, describes how to develop aCCFs from the CCFs data set and why ATR20 was selected (e.g. Section 2.3 of van Manen and Grewe, 2019).

References:
Fuglestvedt, J. S., Shine, K. P., Berntsen, T., Cook, J., Lee, D. S., Stenke, A., Skeie, R. B., Velders, G. J. M., and Waitz, I. A.: Transport impacts on atmosphere and climate: metrics, Atmospheric Environment, 44, 4648–4677, https://doi.org/10.1016/j.atmosenv.2009.04.044, 2010.

Grewe, V., and Dahlmann, K.: How ambiguous are climate metrics? And are we prepared to assess and compare the climate impact of new air traffic technologies?, Atmospheric Environment, 106, 373–374, https://doi.org/10.1016/j.atmosenv.2015.02.039, 2015.

Grewe, V., Champougny, T., Matthes, S., Frömming, C., Brinkop, S., Søvde, O. A., Irvine, E. A., and Halscheidt, L.: Reduction of the air traffic's contribution to climate change: A REACT4C case study, Atmospheric Environment, 94, 616–625, https://doi.org/10.1016/j.atmosenv.2014.05.059, 2014.

Van Manen, J. and Grewe, V.: Algorithmic climate change functions for the use in eco-efficient flight planning, Transportation Research Part D: Transport and Environment, 67, 388–405, https://doi.org/10.1016/j.trd.2018.12.016, 2019.

- **2)** [Page 8, Section 2.5.4 and Appendix A] While the methodology selected to model the contrail climate impact was commonly used in previous studies, its limitations should be acknowledged and discussed in the paper. An optimization algorithm based on the contrail length might be overly simplistic because it does not account for differences in the contrail radiative forcing, lifetime and coverage area:

Reply: Thank you very much. We acknowledge that this routing option is the simple option for contrail avoidance and has limitations. Thus, we modified Section 2.5.4 to describe more details of what this routing option minimizes, and to clarify its limitations: on page 8 line 9, "… developed the routing option  **to avoid contrail formations by using the submodel CONTRAIL (version 1.0; Frömming et al., 2014), which calculates the potential persistent contrail cirrus coverage Potcov (Ponater et al., 2002; Burkhardt et al., 2008; Burkhardt and Kärcher, 2009; Grewe et al. 2014b) within an EMAC grid box.**  **The Potcov represents the fraction of the grid box, which can be maximally covered by contrails under the simulated atmospheric condition. The threshold for contrail formation is determined from a parameterization scheme based on the thermodynamic theory of contrails, i.e., the Schmidt-Appleman theory (Schmidt, 1941; Appleman, 1953; Schumann, 1996). In the CONTRAIL submodel, Potcov indicates the difference between the maximum possible coverage of both, contrails and cirrus, and the coverage of natural cirrus alone; values of Potcov along the waypoints are taken from the nearest grid box (Table 2). With that, we define a contrail distance ($PCC_{dist}$) in km(contrail) as Potcov multiplied by the flight distance in km. The corresponding routing option minimizes the total contrail distance of a flight and thus the** objective function  **is formulated as:**…."

In addition, we added the text to the same Section 2.5.4: on page 8 line 19, "**Note that the objective function**

**is formulated in the simple form to consider only the contrail distance. Thus, further physical processes such as contrail spreading, changes in contrail coverage area, contrail lifetime, and the contrail radiative forcing are not included**."

We believe that this contrail routing option (using $PCC_{dist}$) is one of the important routing options to study characteristics of aircraft routing strategies regarding contrails. AirTraf 2.0 includes the climate routing option, in which contrail effects are included in a different way (using $ATR20_{contrail}$). These two options work differently on contrail avoidance in flight trajectory optimizations; this interesting aspect is additionally discussed (please see the reply to the referee major comment (4)).

This modification is related to our reply to the comment (20) of referee #1.

- Firstly, Eq. (A5) assumes that contrails always cool during the day because it has a negative $aCCF_{contrail}$ and $ATR20_{contrail}$. However, this is not true as many other studies have shown that contrails can either warm or cool during the day, depending on meteorology (such as ambient cirrus), radiation, and the solar zenith angle.

  Reply: The $aCCF_{contrail}$ for the day-time contrails, which is defined in Eq. (A5) on page 15 in the Appendix, can take positive and negative values, depending on the outgoing long-wave radiation (OLR) (the threshold is $-193.18\,\mathrm{Wm^{-2}}$).

  Through the derivation process of $aCCF_{contrail}$, we have investigated the relationships between net RF for the day-time contrails and the relevant meteorological variables. The results showed the highest correlation with OLR ($R = 0.86$), whereas the introduction of a second parameter, such as temperature and solar zenith angle, did not improve the correlation. Related to this, we are preparing another manuscript for Geoscientific Model Development, which is the model description paper on the submodel ACCF and describes the derivation of $aCCF_{contrail}$ in detail; we refer to it as "Yin et al. (manuscript in preparation, 2019)," e.g. on page 9 line 18, and on page 15 line 14.

  To make clear the point which the referee noted, we added the text: on page 16 line 6 in the Appendix, "… calculated by AirClim. **The $aCCF_{contrail}$ for the night-time contrails takes positive values; if the temperature is less than 201 K, $aCCF_{contrail}$ for the night-time contrails is set to zero. The $aCCF_{contrail}$ for the day-time contrails can take positive and negative values, depending on the OLR (the threshold is $-193.18\,\mathrm{Wm^{-2}}$).**"

  This modification is related to our reply to the comment (22) of referee #1.

- Secondly, some contrails formed during the day could also have lifetimes of up to 19 hours [Ref.2] and persist through the night, subsequently turning to a warming contrail, but the methodology does not appear to have considered the contrail lifetime. In Eq. (A5), it is also unclear on the conditions/time boundaries which constitutes as day-time and night-time.

  Reply: Thank you very much. To reply to this referee comment and to the next referee comment, let us explain the derivation of $aCCF_{contrail}$ briefly. First, the contrail RF data set was calculated following these steps:

  (a) Lagrangian trajectories (air parcels) were computed by using the ERA-Interim reanalysis data (the methodology is described by Irvine et al., 2014); the trajectories were initialized over the north Atlantic (1 degree horizontal spacing) at three vertical levels (300, 250 and 200 hPa) in winters of 1994, 1995 and 2003. The contrail lifetime was calculated by analyzing each of the trajectories to see how long the conditions persisted for: relative humidity with respect to ice above 98 % and a temperature below 235 K.

  (b) Contrail properties were calculated along the trajectories by following Schumann et al. (2017), where an effective radius for contrail cirrus ice particle was set to 23 microns described by Schumann et al. (2011). The contrail optical depth was calculated by a simple formula for the extinction coefficient (Unterstrasser

and Gierens, 2010), where the initial contrail depth was set to 200 m (Grewe et al., 2014).

(c) The long-wave and short-wave RFs were calculated from the trajectory data by using the parametric equations described by Schumann et al. (2012). The area covered by each contrail was assumed constant along the trajectory. By taking values from Grewe et al. (2014), we used a contrail width of 200 m, and a contrail length of the square root of the grid box area (1 degree by 1 degree grid). The net RF was calculated for each contrail and was converted to a global-mean value by following Grewe et al. (2014). The contrail RF data set was obtained, in which the lifetime of contrails ranges from 3 to 48 hours.

The aCCF$_{contrail}$ was derived based on regression analyses for the RF data set. The methodology was based on that used by van Manen and Grewe (2019) to derive the other aCCFs for ozone, methane and water vapour. For the regression analyses, a constraint on deriving aCCF$_{contrail}$ was that only meteorological information available at the time of flight can be used. In addition, we restricted the calculation to conventional meteorological data, so that aCCF$_{contrail}$ was simple to implement. This means, for example, no information on the contrail lifetime could be used, because this is not something which can be estimated a priori from meteorological data. Since a lifetime was required to be input to the net RF calculations, we chose a contrail lifetime of six hours for all contrails, because 92 % of contrails have a lifetime up to six hours in the data set. Night-time and day-time contrails were analyzed separately. The night-time contrails referred to contrails with their entire (six hours) lifetime occurring at night; the day-time contrails referred to contrails which existed only during daylight hours and those which had part of their lifetime during the day. The obtained aCCF$_{contrail}$ (Eq. (A5) on page 15 in the Appendix) was converted from RF to ATR20 by multiplying a factor of 0.114 (provided by Katrin Dahlmann, DLR).

The derived aCCF$_{contrail}$ has been assessed by plotting the original net RF with the RF calculated by using aCCF$_{contrail}$. In addition, the performance of aCCF$_{contrail}$ has been assessed against the rest of the contrails with lifetimes of 3 to 48 hours in the data set. The Spearman's rank correlation coefficient and the ability to correctly predict the sign of the forcing were examined. For day-time contrails with lifetimes of 6 hours, for example, the coefficient was R = 0.86, and the ability (in percentage) was 88 %; for those with lifetimes of 12 hours, R = 0.83 and 78 % were obtained. These results provide the confidence in the use of aCCF$_{contrail}$ (Eq. (A5)) in the aircraft routing decision.

As for the conditions/time boundaries, the procedure of calculating aCCF$_{contrail}$ is as follows: for locations where contrails could form (potcov > 0), the local time and solar zenith angle are calculated. If the contrail forms in darkness, the time of sunrise is then calculated. If the time between the local time and sunrise is greater than six hours, the night-time aCCF$_{contrail}$ is applied. If the contrail forms in daylight, or in darkness but with less than six hours before sunrise, the day-time aCCF$_{contrail}$ is applied. These calculations are implemented online in EMAC by another submodel named ACCF. The derivation of aCCF$_{contrail}$ described above and details of the submodel ACCF will be described in the forthcoming paper of Yin et al. (manuscript in preparation, 2019), which is referred in the present manuscript, e.g. on page 9 line 18, and on page 15 line 14.

We believe that the conditions/time boundaries are useful information for readers. Thus, we added the text: on page 16 line 6 in the Appendix, "**As for the time boundaries of day and night, the local time and solar zenith angle are calculated for locations where contrails could form (potcov > 0). For locations in darkness, the time of sunrise is then calculated. If the time between the local time and sunrise is greater than six hours, the aCCF$_{contrail}$ for the night-time contrails is applied. If the contrail forms in daylight, or in darkness but with less than six hours before sunrise, the aCCF$_{contrail}$ for the day-time contrails is applied. These calculations are performed online in EMAC by the submodel ACCF.**"

References:
Grewe, V., Frömming, C., Matthes, S., Brinkop, S., Ponater, M., Dietmüller, S., Jöckel, P., Garny, H., Tsati, E., Dahlmann, K., et al.: Aircraft routing with minimal climate impact: the REACT4C climate cost function modelling approach (V1.0), Geoscientific Model Development, 7, 175–201, https://doi.org/10.5194/gmd-7-

175-2014, 2014.

Irvine, E. A., Hoskins, B. J., Shine, K. P.: A Lagrangian analysis of ice-supersaturated air over the North Atlantic, J. Geophys. Res., 119, 1, 90–100, https://doi.org/10.1002/2013JD020251, 2014.

Schumann, U., Baumann, R., Baumgardner, D., Bedka, S. T., Duda, D. P., Freudenthaler, V., Gayet, J.-F., Heymsfield, A. J., Minnis, P., Quante, M., Raschke, E., Schlager, H., Vázquez-Navarro, M., Voigt, C., and Wang, Z.: Properties of individual contrails: a compilation of observations and some comparisons, Atmos. Chem. Phys., 17, 403–438, https://doi.org/10.5194/acp-17-403-2017, 2017.

Schumann, U., Mayer, B., Graf, K., and Mannstein, H.: A Parametric Radiative Forcing Model for Contrail Cirrus, Journal of Applied Meteorology and Climatology, 51, 6, https://doi.org/10.1175/JAMC-D-11-0242.1, 1391–1406, 2012.

Schumann, U., Mayer, B., Gierens, K., Unterstrasser, S., Jessberger, P., Petzold, A., Voigt, C., and Gayet, J-F.: Effective Radius of Ice Particles in Cirrus and Contrails, Journal of the Atmospheric Sciences, 68, 2, 300–321, https://doi.org/10.1175/2010JAS3562.1, 2011.

Unterstrasser, S. and Gierens, K.: Numerical simulations of contrail-to-cirrus transition – Part 1: An extensive parametric study, Atmos. Chem. Phys., 10, 2017–2036, https://doi.org/10.5194/acp-10-2017-2010, 2010.

Van Manen, J. and Grewe, V.: Algorithmic climate change functions for the use in eco-efficient flight planning, Transportation Research Part D: Transport and Environment, 67, 388–405, https://doi.org/10.1016/j.trd.2018.12.016, 2019.

- Thirdly, the $ATR20_{contrail}$ could also be influenced by contrail spreading and its coverage area. However, Eq. (A5) and "$ATR20_{contrail} = aCCF_{contrail}$ x $PCC_{dist}$" does not account for the change in contrail coverage area. Further clarification on these aspects are required.

  Reply: To calculate the contrail RF data set, which was used to derive $aCCF_{contrail}$, the area covered by each contrail was assumed constant along the contrail trajectory. By taking values from Grewe et al. (2014), we used a contrail width of 200 m, and a contrail length of the square root of the grid box area (1 degree by 1 degree grid). We believe that the replies to the referee major comment (2) (starting with "While the methodology") and to the referee major comment mentioned above (starting with "Secondly, some contrails") answer this referee comment. More details will be described in forthcoming paper of Yin et al. (manuscript in preparation, 2019), which is the model description paper on the submodel ACCF.

  Reference:
  Grewe, V., Frömming, C., Matthes, S., Brinkop, S., Ponater, M., Dietmüller, S., Jöckel, P., Garny, H., Tsati, E., Dahlmann, K., et al.: Aircraft routing with minimal climate impact: the REACT4C climate cost function modelling approach (V1.0), Geoscientific Model Development, 7, 175–201, https://doi.org/10.5194/gmd-7-175-2014, 2014.

- **3)** [Page 10, Lines 17 to 19] The results and Figure 2 show that flight trajectories based on the cash operating cost (COC) optimization and minimum fuel consumption always selects a higher cruising altitude. However, this is very likely due to limitations of BADA 3. According to Nuic et al.[Ref.3], BADA 3 has a tendency to underestimate aircraft fuel consumption at higher altitudes and Mach numbers as the compressibility effect and wave drag are not modelled. While I understand that a more accurate version of BADA (BADA 4) is available, obtaining access to it can be challenging. Despite this, the authors should include more discussion on the effects of BADA 3 on their results, as well as acknowledge the limitations of BADA 3.

  Reply: Thank you very much. We added the new figure "Figure 3" and rewrote the text to discuss the effects of BADA 3 and its limitations: on page 10 line 18, "… flight altitudes (∼FL410, 12.5 km). **Figure 3 shows**

**the mean fuel consumption (in kg(fuel)/min$^{-1}$) vs. mean flight altitude (in km) for individual flights for the three routing options.** Because fuel consumption decreases  **as a result of** aerodynamic drag reduction **at high altitudes** (Fichter et al., 2005; Schumann et al., 2011; Yamashita et al., 2016)**, the COC optimum trajectories select the high flight altitudes, as shown in Fig. 3. We acknowledge that limitations of BADA 3 affect the selection of the flight altitudes (the same applies to the fuel, the NO$_x$, the H$_2$O and the SOC options; see Fig. S3 in the Supplementary material). According to Nuic et al. (2010), BADA 3 has a tendency to underestimate aircraft fuel consumption at high altitudes and Mach numbers, as the compressibility effect and wave drag are not modeled. These effects will cause differences in the selection of the flight altitudes.**"

Related to this, we added the paper to the References:
− On page 20 line 10, "**Nuic, A., Poles, D., Mouillet, V.: BADA: An advanced aircraft performance model for present and future ATM systems, Int. J. Adapt. Control Signal Process., 24, 10, 850−866, https://doi.org/10.1002/acs.1176, 2010.**"

As we add the new figure, we changed the original figure number: on page 10 line 21, "Figure 4 shows …"; on page 10 line 25, "… it is apparent from Fig. 4 …"; and on page 24 in the caption, "Figure 4."

[Figure]

**Figure 3. Mean fuel consumption vs. mean flight altitude for 103 individual flights obtained by the contrail formation, the COC and the climate impact routing options.**

- **4)** [Page 11, Lines 9 to 11] "The contrail option shows the minimum contrail distance and decreases ATR20$_{total}$… This option allows aircraft to widely detour the potential contrail regions". This sentence requires further clarification: given that the authors mentioned in Page 9 Line 26 that the "ATR20$_{contrail}$ can take positive and negative values, because of the day-time and night-time contrail effects", it should be made clear in the discussion on if the algorithm: (i) actively forms cooling contrails during the day and avoids forming warming contrails during the night; or (ii) minimises the overall contrail length at all times.

Reply: Thank you very much. We added the new figure "Figure 5" and rewrote the text to make clear how the algorithm works on contrail avoidance: on page 11 line 11, "… is used in Eqs. (1) and (5)**; see below for more discussion**)"; and on page 11 line 24, "**Figure 5 shows the contrail distance (in ×10$^3$ km) vs. ATR20$_{contrail}$ (in ×10$^{-7}$ K) for individual flights for the contrail, the COC, and the climate options. We see that the contrail option decreases the contrail distance drastically and shows the positive values of ATR20$_{contrail}$ for almost all the flights. On the other hand, the climate option has the longer contrail distances than those of the contrail option (although the climate option achieves the second-shortest total contrail distance, as shown in Table 4) and shows the negative values of ATR20$_{contrail}$ for many**

**flights. These results imply that the contrail option minimizes the overall contrail distance at all times, whereas the climate option actively forms cooling contrails during the day and avoids the formation of warming contrails during the day and night.**"

[Figure]

**Figure 5. Contrail distance vs. ATR20$_{contrail}$ for 103 individual flights obtained by the contrail formation, the COC and the climate impact routing options.**

- **Minor Comments:**
  **1)** [Page 1, Line 22] Replace "non-volatile black carbon (BC or soot)" with "non-volatile particulate matter such as BC" for correctness in terminology [Ref.4]. This is because black carbon (BC) is a subset of non-volatile particulate matter (nvPM), while the term "soot" includes both nvPM (BC and metallic compounds) and organic compounds.

  Reply: Thank you very much. We rewrote the word: on page 1 line 22, "…  **non-volatile particulate matter such as black carbon (BC)**…."

- **2)** [Page 2, Line 7] The sentence, "The emitted $CO_2$ has a long residence time (a century)", should be corrected. According to Joos et al.1, however, the emitted $CO_2$ can remain in the atmosphere after a millennium.

  Reply: Thank you very much. In this sentence, we would like to note a long-lasting $CO_2$ impact in the atmosphere. Thus, we modified the sentence: on page 2 line 6, " **the emitted $CO_2$ becomes uniformly mixed in the whole atmosphere and its perturbation remains for millennia.**" This modification is related to our reply to the comment (9) of the referee #1.

- **3)** [Page 3, Line 15] Remove "the" from this sentence "the today's aircraft routing focuses on the minimum economic cost".

  Reply: We removed the word "the" from the sentence: on page 3 line 15, "…  today's aircraft routing focuses on  minimum economic cost…."

- **4)** [Page 8, Line 22] There appears to be inconsistencies in the acronyms: $C_{time}$ and $C_{fuel}$ was used in line 22. However, $c_t$ and $c_f$ are used in Eq. (6). This can confuse future readers.

  Reply: Thank you very much. On page 8 line 22, we explain the definition of the cost index: Cost Index (CI)

= time cost / fuel cost, where CI is a kind of dimensionless coefficient showing the ratio of time cost [USDollar] to fuel cost [USDollar]; and $C_{time}$ and $C_{fuel}$ represent the "time cost" and the "fuel cost" of the definition. On the other hand, $c_t$ and $c_f$ used in Eq. (6) are the unit costs of time [USDollar/s] and fuel [USDollar/kg(fuel)], respectively, as listed in Table 1. Thus, $C_{time}$ and $C_{fuel}$ are different from $c_t$ and $c_f$, respectively.

We agree that those parameters can confuse readers, and thus we rewrote the sentence: on page 8 line 22, "(CI = $C_{time}$ **time cost** / $C_{fuel}$ **fuel cost**, where $C$ denotes a cost)." Related to this, we moved the phrase "where $C$ denotes a cost" from the current position to the new position: on page 9 lines 13-14, "$f$ = COC = …. + $C_{engine}$, **where $C$ denotes a cost. A detailed….**"

- **5)** [Section 2.5.7] Please acknowledge the large uncertainties in the global temperature response, especially from contrails (ATR20$_{contrail}$) due to uncertainties in the contrail efficacy [Ref.5,6].

  Reply: We added the sentence to acknowledge the point: on page 9 line 26, "… ATR20$_{contrail,i}$ can take positive and negative values, because  **the aCCF$_{contrail}$ consists of two formulas for** the day-time and night-time contrail effects **(see Eqs. (12) and (A5) in the Appendix). We acknowledge the large uncertainties in the global temperature response, especially from contrails (ATR20$_{contrail}$) due to uncertainties in the efficacy of the contrail forcing (Hansen et al., 2005; Ponater et al., 2005). In addition,** the aCCFs are derived…." The rewriting highlighted by blue texts comes from our reply to the comment (22) of the referee #1.

  Related to this, we added the two papers to the References:
  − On page 19 line 4, "**Hansen, J., Sato, M., Ruedy, R., Nazarenko, L., Lacis, A., Schmidt, GA., Russell, G., Aleinov, I., Bauer, M., Bauer, S., Bell, N., Cairns, B., Canuto, V., Chandler, M., Cheng, Y., Genio, A Del., Faluvegi, G., Fleming, E., Friend, A., et al.: Efficacy of climate forcings, Journal of Geophysical Research: Atmospheres, 110, D18104, 1−45, https://doi.org/10.1029/2005JD005776, 2005.**"
  − On page 20 line 15, "**Ponater, M., Marquart, S., Sausen, R., Schumann, U.: On contrail climate sensitivity, Geophysical Research Letters, 32, L10706, 1–5, https://doi.org/10.1029/2005GL022580, 2005.**"

- **6)** [Section 4: Discussion] The authors should highlight that these results (climate benefits) is likely an upper limit, because airspace congestion and air traffic management could minimise the flexibility for flights to perform these trajectory optimisations.

  Reply: We agree with the referee comment. We rewrote the text: on page 11 line 24, "… on December 1, 2015, as shown in Table 3). **Finally, we believe that the climate benefits described above are most likely an upper limit, because airspace congestion and air traffic management could reduce the flexibility for flights to perform these trajectory optimizations.**" Similarly, we added the sentence to the conclusions to highlight the same point: on page 15 line 1, "… performance than the contrail option. **We believe that these climate benefits are most likely an upper limit.** The simulation results were…."

- **7)** [Eq. (5) and Table 1] Consider using different notations for the mass fuel flow rate ($f_{ref}$), as this is similar to the objective function ($f$) and can lead to confusion.

  Reply: We understood the referee comment, and we reconsidered to change the notation of $f_{ref}$. Nevertheless, we are hesitating to change it, because "$f_{ref}$" has been already used from the previous version of AirTraf 1.0 (Sect. 2.3, Sect. 2.6 and Table 1 of Yamashita et al., 2016). For the sake of consistency, it could be better for readers to use the same notation. Nevertheless, if this change is essential for the revision of the manuscript, we will follow the suggestion by the referee #2 and change the notation, if the editor decides to do so.

  Reference:
  Yamashita, H., Grewe, V., Jöckel, P., Linke, F., Schaefer, M., and Sasaki, D.: Air traffic simulation in

chemistry-climate model EMAC 2.41: AirTraf 1.0, Geoscientific Model Development, 9, 3363–3392, https://doi.org/10.5194/gmd-9-3363-2016, 2016.

- **8)** [Appendix A, Eq. (A5)] $aCCF_{contrail} = \ldots$, if Potcov $\geq 0$: should this be $> 0$ instead? Similarly, $aCCF_{contrail} = 0$, if Potcov $< 0$: should this be $\leq 0$ instead?

  Reply: Thank you very much. We corrected the equations: on page 15 Appendix A, Eq. (A5), $aCCF_{contrail} = \ldots$, if Potcov $\geq > 0$ .and. nighttime; $aCCF_{contrail} = \ldots$, if Potcov $\geq > 0$ .and. daytime; $aCCF_{contrail} = 0$, if Potcov $< \leq 0$.

**References provided by the referee #2**
1. Joos F, Roth R, Fuglestvedt JS, Peters GP, Enting IG, von Bloh W, Brovkin V, Burke EJ, Eby M, Edwards NR, Friedrich T, Frölicher TL, Halloran PR, Holden PB, Jones C, Kleinen T, Mackenzie FT, Matsumoto K, Meinshausen M, et al. Carbon dioxide and climate impulse response functions for the computation of greenhouse gas metrics: a multi-model analysis. Atmos Chem Phys. 2013;13(5):2793-2825. doi:10.5194/acp-13-2793-2013.

2. Haywood JM, Allan RP, Bornemann J, Forster PM, Francis PN, Milton S, Rädel G, Rap A, Shine KP, Thorpe R. A case study of the radiative forcing of persistent contrails evolving into contrail-induced cirrus. J Geophys Res Atmos. 2009;114(D24201). doi:doi:10.1029/2009JD012650.

3. Nuic A, Poles D, Mouillet V. BADA: An advanced aircraft performance model for present and future ATM systems. Int J Adapt Control Signal Process. 2010;24(10):850-866. doi:10.1002/acs.1176.

4. Petzold A, Ogren JA, Fiebig M, Laj P, Li S-M, Baltensperger U, Holzer-Popp T, Kinne S, Pappalardo G, Sugimoto N. Recommendations for reporting "black carbon" measurements. Atmos Chem Phys. 2013;13(16):8365-8379.

5. Hansen J, Sato M, Ruedy R, Nazarenko L, Lacis A, Schmidt GA, Russell G, Aleinov I, Bauer M, Bauer S, Bell N, Cairns B, Canuto V, Chandler M, Cheng Y, Genio A Del, Faluvegi G, Fleming E, Friend A, et al. Efficacy of climate forcings. J Geophys Res. 2005;110(D18):D18104. doi:10.1029/2005JD005776.

6. Ponater M, Marquart S, Sausen R, Schumann U. On contrail climate sensitivity. Geophys Res Lett. 2005;32(10).